# A fungal member of the *Arabidopsis thaliana* phyllosphere antagonizes *Albugo laibachii* via a GH25 lysozyme

**Katharina Eitzen[1,2], Priyamedha Sengupta[1], Samuel Kroll[2], Eric Kemen[2,3]\*, Gunther Doehlemann[1]\***

[1]Institute for Plant Sciences and Cluster of Excellence on Plant Sciences (CEPLAS), University of Cologne, Center for Molecular Biosciences, Cologne, Germany; [2]Max Planck Institute for Plant Breeding Research, Cologne, Germany; [3]Department of Microbial Interactions, IMIT/ZMBP, University of Tübingen, Tübingen, Germany

**Abstract** Plants are not only challenged by pathogenic organisms but also colonized by commensal microbes. The network of interactions these microbes establish with their host and among each other is suggested to contribute to the immune responses of plants against pathogens. In wild *Arabidopsis thaliana* populations, the oomycete pathogen *Albugo laibachii* plays an influential role in structuring the leaf phyllosphere. We show that the epiphytic yeast *Moesziomyces bullatus* ex *Albugo* on Arabidopsis, a close relative of pathogenic smut fungi, is an antagonistic member of the *A. thaliana* phyllosphere, which reduces infection of *A. thaliana* by *A. laibachii*. Combination of transcriptomics, reverse genetics, and protein characterization identified a GH25 hydrolase with lysozyme activity as a major effector of this microbial antagonism. Our findings broaden the understanding of microbial interactions within the phyllosphere, provide insights into the evolution of epiphytic basidiomycete yeasts, and pave the way for novel biocontrol strategies.

**\*For correspondence:**
eric.kemen@uni-tuebingen.de (EK);
g.doehlemann@uni-koeln.de (GD)

**Competing interests:** The authors declare that no competing interests exist.

## Introduction

Plants are colonized by a wide range of microorganisms. While some microbes enter the plant and establish endophytic interactions with a broad range of outcomes from beneficial to pathogenic, plant surfaces harbor a large variety of microbial organisms. Recent research has focused largely on the importance of the rhizosphere microbiota in nutrient acquisition, protection from pathogens, and boosting overall plant growth and development (*Ritpitakphong et al., 2016*; *Walker et al., 2003*; *Bulgarelli et al., 2013*). However, the above ground parts of the plant including the phyllosphere are colonized by diverse groups of microbes that also assist in plant protection and immunity (*Busby et al., 2016*; *Mikiciński et al., 2016*). The environment has a major impact on the microbial communities of the leaf surface, ultimately influencing their interactions with the host (*Stone et al., 2018*).

Scale-free network analysis was performed with the leaf microbial population of *Arabidopsis thaliana* (*Agler et al., 2016*). The majority of the interactions between kingdoms, e.g. fungi and bacteria, were found to be negative, consistent with the fact that rather the antagonistic interactions stabilize a microbial community (*Coyte et al., 2015*). Phyllosphere network analysis of *A. thaliana* identified a small number of microbes as 'hub' organisms, i.e. influential microbes that have severe effects on the community structure. The major hub microbe in the *A. thaliana* phyllosphere is the oomycete *Albugo laibachii*, which is a pathogenic symbiont biotrophic of Arabidopsis (*Agler et al., 2016*). This pathogen has been shown to significantly reduce the bacterial diversity of epiphytic and endophytic leaf habitats. Since bacteria generally comprise a large proportion of the phyllosphere microbiome

**eLife digest** Much like the 'good bacteria' that live in our guts, many microscopic organisms can co-exist with and even benefit the plants they live on. For instance, the yeast *Moesziomyces bullatus ex Albugo* (MbA for short) can shield the leaves of its plant host against white rust, a disease caused by the organism *Albugo laibachii*. Studies have started to unveil how the various microbes at the surface of leaves interact and regulate their own community, yet the genetic mechanisms at play are less well-known.

To investigate these processes, Eitzen et al. examined the genes that were switched on when MbA cells were in contact with *A. laibachii* on a leaf. This experiment revealed a few gene candidates that were then deleted, one by one, in MbA cells. As a result, a gene emerged as being key to protect the plant from white rust. It produces an enzyme known as the GH25 hydrolase, which, when purified, could reduce *A. laibachii* infections on plant leaves.

Bacteria, fungi and other related microorganisms cause many diseases which, like white rust, can severely affect crops. Chemical methods exist to prevent these infections but they can have many biological and ecological side effects. A solution inspired by natural interactions may be safer and more effective at managing plant diseases that affect valuable crops. Harnessing the interactions between microbes living on plants, and the GH25 enzyme, may offer better disease control.

(*Vorholt, 2012*), phylogenetic profiling of *A. thaliana* was also directed toward identifying a small group of bacteria that frequently colonize *A. thaliana* leaves. The analysis helped to develop a synthetic community of bacteria for experiments in gnotobiotic plants.

Besides bacteria and oomycetes, the microbiota of the *A. thaliana* leaf also comprises a broad range of fungi. Among those fungi, basidiomycete yeasts are frequently found and the most frequent ones are the epiphytic basidiomycete genus Dioszegia (*Agler et al., 2016*), as well as an anamorphic yeast associated with *A. laibachii* infection belonging to the Ustilaginales. This order includes many pathogens of important crop plants, for example corn smut and loose smut of oats, barley, and wheat are caused by *Ustilago maydis*, *U. avenae*, *U. nuda*, and *U. tritici*, respectively. Generally, the pathogenic development of smut fungi is linked with sexual recombination and plant infection is only initiated upon mating, when two haploid sporidia form a dikaryotic filament (*Brefort et al., 2009*). Ustilaginales *Pseudozyma* sp. yeasts, however, are found mostly in their anamorphic stage in nature. They tend to epiphytically colonize a wide range of habitats, where an infrequent sexual recombination might occur when they meet on a susceptible host (*Kruse et al., 2017*). Phylogenetic reconstruction (*Kruse et al., 2017*; *Wang et al., 2015*) showed that the smut pathogen of millet, *Moesziomyces bullatus* and four species of Pseudozyma, namely *P. antarctica*, *P. aphidis*, *P. parantarctica*, and *P. rugulosa* form a monophyletic group. The latter do represent anamorphic and culturable stages of *M. bullatus* and, hence, can be grouped to this genus. Moesziomyces strains have been reported in a number of cases to act as microbial antagonists. A strain formerly classified as *Pseudozyma aphidis* (now *M. bullatus*) inhibited *Xanthomonas campestris* pv. *vesicatoria*, *X. campestris* pv. *campestris*, *Pseudomonas syringae* pv. *tomato*, *Clavibacter michiganensis*, *Erwinia amylovora*, and *Agrobacterium tumefaciens* in vitro and also led to the activation of induced defense responses in tomato against the pathogen (*Barda et al., 2015*). It was reported that *P. aphidis* can parasitize the hyphae and spores of *Podosphaera xanthii* (*Gafni et al., 2015*). *Pseudozyma churashimaensis* was reported to induce systemic defense in pepper plants against *X. axonopodis*, Cucumber mosaic virus, Pepper mottle virus, Pepper mild mottle virus, and broad bean wilt virus (*Lee et al., 2017*).

In the present study, we explored the antagonistic potential of an anamorphic Ustilaginales yeast within the leaf microbial community of *A. thaliana*. We show that *Moesziomyces bullatus* ex *Albugo* on Arabidopsis (which will be referred to as *MbA* from further on) prevents infection by the oomycete pathogen *A. laibachii* and identified fungal candidate genes that were upregulated in the presence of *A. laibachii*, when both microbes were co-inoculated to the host plant. A knockout mutant of one of the candidates, which belongs to the glycoside hydrolase – family 25 (GH25) – was found to lose its antagonistic activity toward *A. laibachii*, providing mechanistic insights into fungal-

oomycete antagonism within the phyllosphere microbiota. Functional characterization of GH25 will be an important step toward establishing *MbA* as a suitable biocontrol agent.

## Results

In a previous study we isolated a basidiomycetous yeast from *Arabidopsis thaliana* leaves infected with the causal agent of white rust, *Albugo laibachii* (*Agler et al., 2016*). This yeast was tightly associated with *A. laibachii* spore propagation. Even after years of subculturing in the lab and re-inoculation of plants with frozen stocks of *A. laibachii* isolate Nc14, this yeast remained highly abundant in spore isolates. Phylogenetic analyses based on fungal ITS-sequencing identified the yeast as *Pseudozyma* sp. Those yeasts can be found across the family of Ustilaginaceae, being closely related to pathogens of monocots like maize, barley, sugarcane, or sorghum (*Zuo et al., 2019*). Based on phylogenetic similarity to the pathogenic smut *Moesziomyces bullatus* which infects millet, several anamorphic Pseudozyma isolates were suggested to be renamed and grouped to *M. bullatus* (*Wang et al., 2015*). Since the *Pseudozyma* sp. that was isolated from *A. laibachii* spores groups into the same cluster, we classified this newly identified strain as *MbA (Moesziomyces bullatus* ex *Albugo* on Arabidopsis).

Based on the identification of *MbA* as having a significant effect on bacterial diversity in the Arabidopsis phyllosphere, we tested its interaction with 30 bacterial strains from 17 different species of a synthetic bacterial community (SynCom, *Supplementary file 1*) of Arabidopsis leaves in one-to-one plate assays. This experiment identified seven strains being inhibited by Moesziomyces, as indicated by halo formation after 7 days of co-cultivation (*Figure 1—figure supplement 1*). Interestingly, this inhibition was not seen when the pathogenic smut fungus *U. maydis* was co-cultivated with the bacteria, indicating a specific inhibition of the bacteria by *MbA* (*Figure 1—figure supplement 1*).

The primary hub microbe in the Arabidopsis phyllosphere was found to be the pathogenic oomycete *A. laibachii,* which was isolated in direct association with Moesziomyces (*Agler et al., 2016*). To test if both species interfere with each other, we deployed a gnotobiotic plate system and quantified *A. laibachii* infection symptoms on Arabidopsis. In control experiments, spray inoculation of only *A. laibachii* spores on Arabidopsis leaves led to about 33% infected leaves at 14 days post infection (dpi) (*Figure 1*). When the bacterial SynCom was pre-inoculated on leaves 2 days before *A. laibachii* spores a significant reduction of *A. laibachii* infection by about 50% was observed (*Figure 1*). However, if Moesziomyces was pre-inoculated with the bacterial SynCom, *A. laibachii* spore production was almost completely abolished. Similarly, the pre-inoculation of only *MbA* resulted in an almost complete loss of *A. laibachii* infection, independently of the presence of a bacterial community (*Figure 1*). The antagonistic effect of *MbA* toward *A. laibachii* was further confirmed using Trypan blue staining of *A. laibachii* infected *A. thaliana* leaves. *A. laibachii* forms long, branching filaments on Arabidopsis leaves at 15 dpi. Contrarily, in the presence of *MbA*, we observed mostly zoospores forming either no or very short hyphae, while further colonization of the leaf with long, branching was not observed (*Figure 1—figure supplement 2*). Together, our findings demonstrate that *MbA* holds a strong antagonistic activity toward *A. laibachii*, resulting in efficient biocontrol of pathogen infection. Thus, *MbA* is an important member of the *A. thaliana* phyllosphere microbial community, with a strong impact on its quantitative composition. However, despite several reports of the basidiomycete yeasts acting as antagonists, genome information of this group is rather limited. To enable a molecular understanding of how *MbA* acts on other members of the phyllosphere community, *MbA* genome information is required. We therefore sequenced the genome of *MbA* and established molecular tools including a protocol for stable genomic transformation to allow functional genetic approaches.

### The genome of *MbA*

Genome sequence of *MbA* was analyzed by Single Molecule Real-Time sequencing (Pacific Biosciences, Menlo Park, CA), which lead to 69,674 mapped reads with an accuracy of 87.3% and 8596 bp sub-read length. Sequence assembly using the HGAP-pipeline (Pacific Biosciences) resulted in 31 contigs with an $N_{50}$ Contig Length of 705 kb. The total length of all contigs results in a predicted genome size of 18.3 Mb (*Table 1*). Gene prediction for the *MbA* genome with Augustus (*Stanke et al., 2004*) identified 6653 protein coding genes, of which 559 carry a secretion signal.

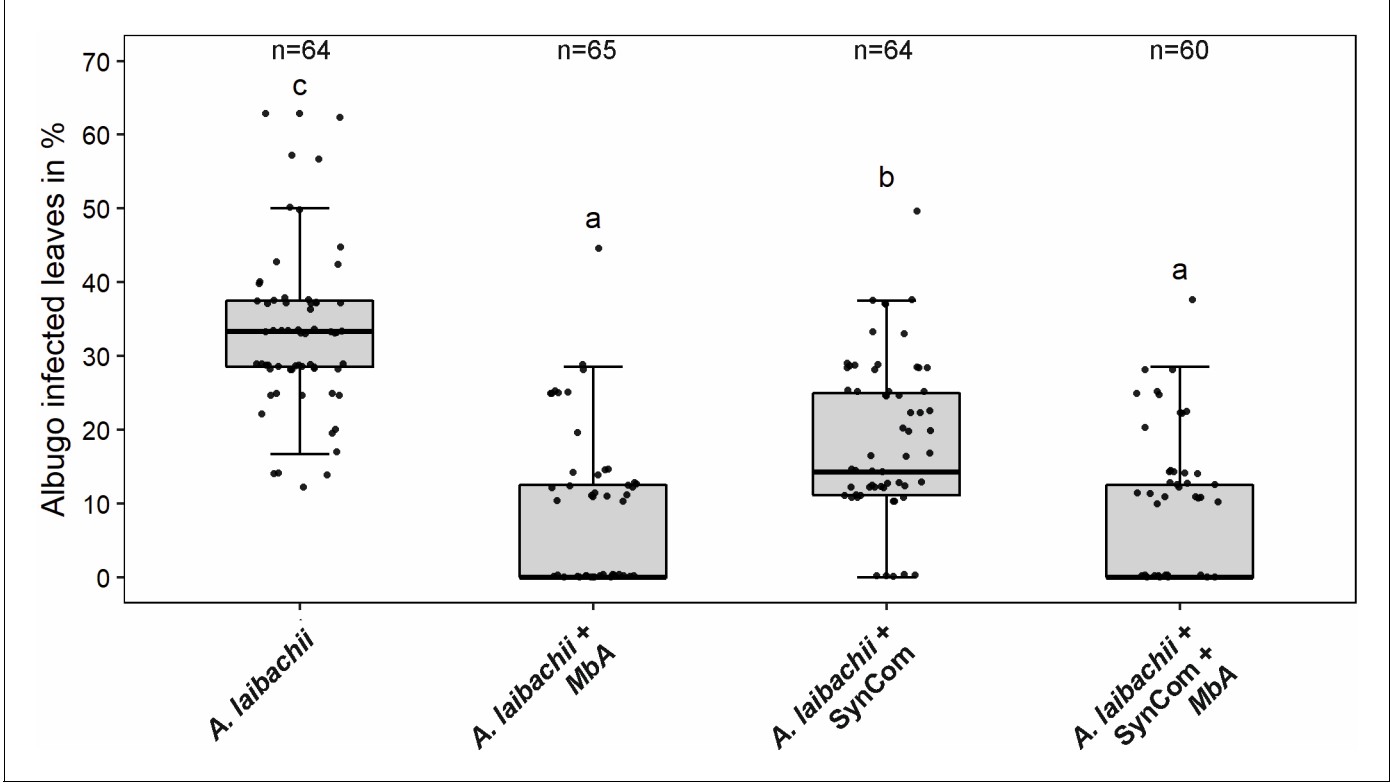

**Figure 1.** Infection assay of *A. laibachii* on *A. thaliana*. Addition of a bacterial SynCom reduces the infection symptoms of *A. laibachii* at 14 days post infection. Those symptoms can be almost abolished by spraying *MbA* to the plant, independently of the presence of the bacterial community. Infections were performed in six individual replicates with 12 technical replicates. N indicates the number of infected plants that were scored for symptoms. An analysis of variance (ANOVA) model was used for pairwise comparison of the conditions, with Tukey's HSD test to determine differences among them. Different letters indicate significant differences (p-values <0.05).

The online version of this article includes the following source data and figure supplement(s) for figure 1:

**Source data 1.** Quantification of *A. laibachii* infection on *A. thaliana*.

**Figure supplement 1.** Biocontrol activity of *MbA*, but not *U. maydis*, against bacterial SynCom members (***Agler et al., 2016***).

**Figure supplement 2.** Trypan blue staining of *A. thaliana* leaves 15 days post infection with *A. laibachii*.

Out of these 559, 380 are predicted to be secreted extracellularly (i.e. they do not carry membrane domains or cell-wall anchors) (***Table 1***). The small genome size and high number of coding genes result in a highly compact genome structure with only small intergenic regions. These are features similarly found in several pathogenic smut fungi such as *U. maydis* and *S. reilianum* (***Table 1***). Remarkably, both *MbA* and *Anthracocystis flocculosa*, which is another anamorphic and apathogenic yeast, show a similarly high rate of introns, while the pathogenic smut fungi have a significantly lower intron frequency (***Table 1***).

To gain better insights in the genome organization of *MbA*, we compared its structure with the *U. maydis* genome, which served as a manually annotated high-quality reference genome for smut fungi (***Kämper et al., 2006***). Out of the 31 *MbA* contigs, 21 show telomeric structures and a high synteny to chromosomes of *U. maydis*, with three of them displaying major events of chromosomal recombination (***Figure 2A***). Interestingly, the Moesziomyces contig 2, on which also homologs to pathogenic loci like the *U. maydis* virulence cluster 2A (***Kämper et al., 2006***) can be found, contains parts of three different *U. maydis* chromosomes (Chr. 2, 5, and 20) (***Figure 2—figure supplement 1***). The second recombination event on contig six affects the *U. maydis* leaf-specific effector protein See1, which is required for tumor formation (***Redkar et al., 2015***). This recombination event is also found in the genome of the maize head smut *S. reilianum*, wherein the *U. maydis* chromosomes 5 and 20 recombined in the promoter region of the *see1* gene (***Figure 2B***). In this respect it should be noted that *S. reilianum*, although infecting the same host, does not produce leaf tumors as *U. maydis* does (***Schirawski et al., 2010***).

**Table 1.** Comparison of genomes and genomic features of known pathogenic and anamorphic Ustilaginomycetes.

| | MbA | U. bromivora | S. scitamineum | S. reilianum | U. maydis | U. hordei | M. pennsylvanicum | A. flocculosa |
|---|---|---|---|---|---|---|---|---|
| Assembly statistics | | | | | | | | |
| Total contig length (Mb) | 18.3 | | 19.5 | 18.2 | 19.7 | 20.7 | 19.2 | 23.2 |
| Total scaffold length (Mb) | | 20.5 | 19.6 | 18.4 | 19.8 | 21.15 | 19.2 | 23.3 |
| Average base coverage | 50× | 154× | 30× | 20× | 10× | 25× | 339× | 28× |
| N50 contig (kb) | 705.1 | | 37.6 | 50.3 | 127.4 | 48.7 | 43.4 | 38.6 |
| N50 scaffold length (kb) | | 877 | 759.2 | 738.5 | 817.8 | 307.7 | 121.7 | 919.9 |
| Chromosomes | 21 | 23 | | 23 | 23 | 23 | | |
| GC-content (%) | 60.9 | 52.4 | 54.4 | 59.7 | 54 | 52 | 50.9 | 65.1 |
| Coding (%) | 62.8 | 54.4 | 57.8 | 62.6 | 56.3 | 54.3 | 54 | 66.3 |
| Coding sequence | | | | | | | | |
| Percentage CDS (%) | 69.5 | 59.8 | 62 | 65.9 | 61.1 | 57.5 | 56.6 | 54.3 |
| Average gene size (bp) | 1935 | 1699 | 1819 | 1858 | 1836 | 1708 | 1734 | 2097 |
| Average gene density (gene/kb) | 0.36 | 0.35 | 0.34 | 0.37 | 0.34 | 0.34 | 0.33 | 0.30 |
| Protein-coding genes | 6653 | 7233 | 6693 | 6648 | 6786 | 7113 | 6279 | 6877 |
| Exons | 11,645 | 11,154 | 10,214 | 9776 | 9783 | 10,907 | 9278 | 19,318 |
| Average exon size (bp) | 1091 | 1101 | 1191 | 1221 | 1230 | 1107 | 527 | 658 |
| Exons/gene | 1.75 | 1.5 | 1.5 | 1.47 | 1.44 | 1.53 | 1.48 | 2.8 |
| tRNA genes | 150 | 133 | 116 | 96 | 111 | 110 | 126 | 176 |
| Noncoding sequence | | | | | | | | |
| Introns | 9333 | 3921 | 3521 | 3103 | 2997 | 3161 | 2999 | 12,427 |
| Introns/gene | 1.40 | 0.54 | 0.53 | 0.47 | 0.44 | 0.44 | 0.48 | 1.81 |
| Average intron length (base) | 163 | 163 | 130.1 | 144 | 142 | 141 | 191.4 | 141 |
| Average intergenic distance (bp) | 769 | 1054 | 1114 | 929 | 1127 | 1186 | 1328 | 1273 |
| Secretome | | | | | | | | |
| Protein with signal peptide | 559 | | 622 | 632 | 625 | 538 | 419 | 622 |
| Secreted without TMD | 380 | | | | 467 | | | 737 |
| – with known domain | 260 | | | | 264 | | | 554 |

[**Brefort et al., 2014**; **Nadal and Gold, 2010**].

Also the third major recombination event, affecting *MbA* contig 8, changes the genomic context genes encoding essential virulence factors in *U. maydis* (*stp1* and *pit1/2*), as well as the A mating type locus, which is important for pheromone perception and recognition of mating partners (**Bölker et al., 1992**). Based on the strong antibiotic activities of *MbA,* we mined the genome of *MbA* for the presence of secondary metabolite gene clusters. Using AntiSMASH, we were able to predict 13 of such clusters, of which three can be assigned to terpene synthesis, three contain nonribosomal peptide synthetases, and one cluster has a polyketide synthase as backbone genes (**Figure 3—figure supplement 1**). Interestingly, the secondary metabolite cluster that is involved in the production of the antimicrobial metabolite ustilagic acid in other Ustilaginomycetes is absent in *MbA* (**Figure 3—figure supplement 1B**). On the contrary, we could identify three *MbA*-specific metabolite clusters that could potentially be involved in the antibacterial activity of *MbA* (**Figure 3— figure supplement 1C**).

A previous genome comparison of the related Ustilaginales yeast *A. flocculosa* with *U. maydis* concluded that this anamorphic strain had lost most of its effector genes, reflecting the absence of a pathogenic stage in this organism (**Lefebvre et al., 2013**). In contrast, *MbA* contains 1:1 homologs of several known effectors with a known virulence function in *U. maydis* (**Table 2**). We previously found that *Moesziomyces* sp. possess functional homologues of the *pep1* gene, a core virulence effector of *U. maydis* (**Sharma et al., 2019**), suggesting that such anamorphic yeasts have the

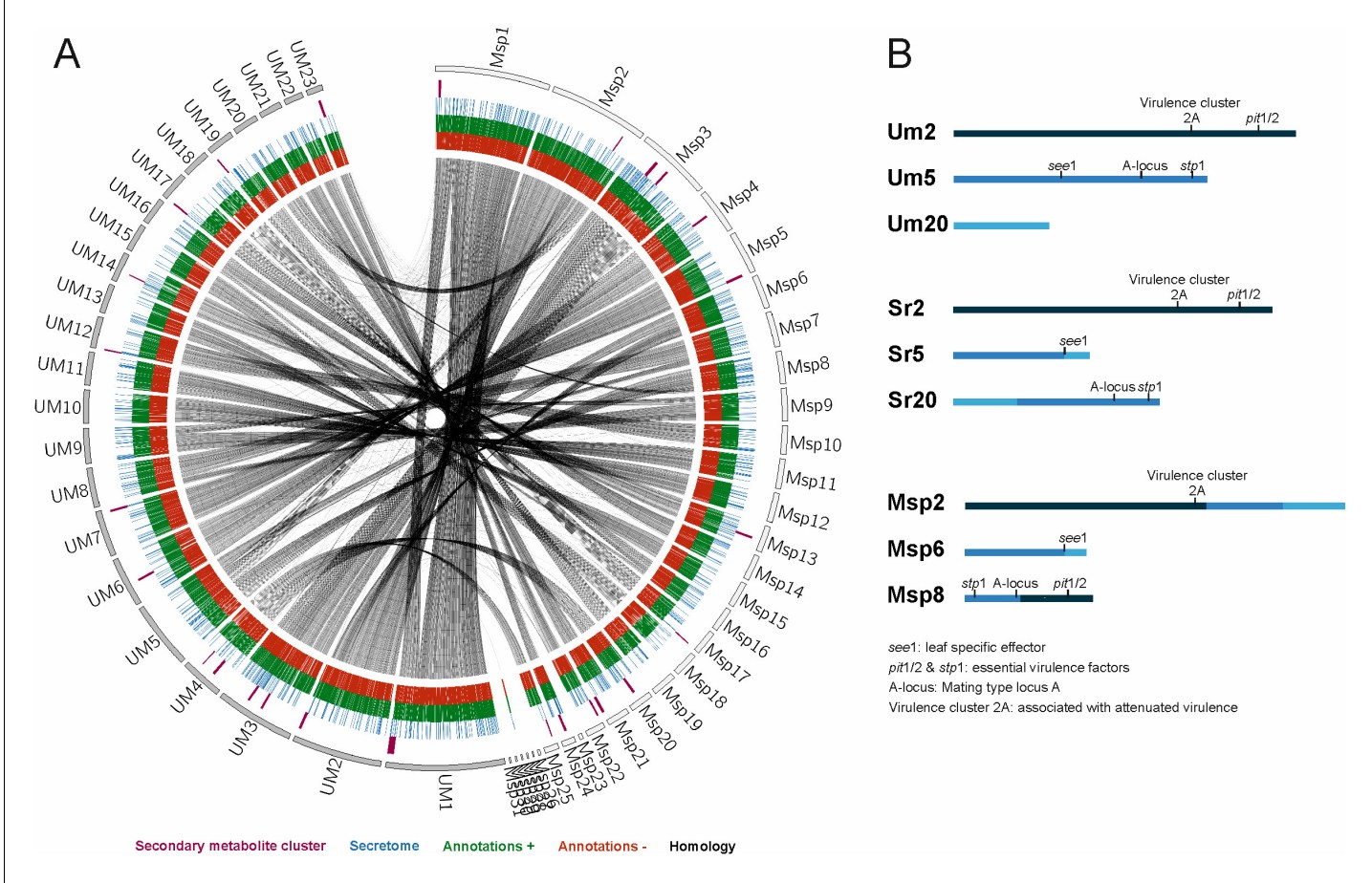

**Figure 2.** Circos comparison of *MbA* and *U. maydis* chromosome structure (**A**). We highlighted potential secondary metabolite clusters, secreted proteins, and gene predictions on both strands (±). (**B**) Homology-based comparisons identified three chromosomal recombination events, which affects the *MbA* contigs 2, 6, and 8.

The online version of this article includes the following figure supplement(s) for figure 2:

**Figure supplement 1.** Genome comparison of *MbA* and *Moesziomyces antarctica* T-34.

potential to form infectious filamentous structures by means of sexual reproduction (*Kruse et al., 2017*). To assess the potential virulence activity of *MbA* effector homologs, we expressed the homolog of the *U. maydis* core effector Pep1 in an *U. maydis pep1* deletion strain (SG200Δ01987). This resulted in complete restoration of *U. maydis* virulence, demonstrating that *MbApep1* encodes a functional effector protein (*Figure 3—figure supplement 2*).

A hallmark of the *U. maydis* genome structure is the presence of large clusters with effector genes, the expression of which is only induced during plant infection (*Kämper et al., 2006*). To assess the presence of potential virulence clusters in *MbA*, we compared all *U. maydis* effector gene clusters to the *MbA* genome, based on homology. This revealed that the 12 major effector clusters of *U. maydis* are present in *MbA*. However, while many of the clustered effector genes are duplicated in pathogenic smut fungi, *MbA* carries only a single copy of each effector gene. This results in the presence of 'short' versions of the *U. maydis* gene effector clusters (*Figure 3—figure supplement 3*). This gets particularly obvious for the biggest and most intensively studied virulence cluster of smut fungi, the effector cluster 19A (*Schirawski et al., 2010*; *Brefort et al., 2014*; *Dutheil et al., 2016*). In *MbA*, only 5 out of the 24 effector genes present in *U. maydis* are conserved in this cluster (*Figure 3*). Interestingly, some anamorphic yeasts like *Kalmanozyma brasiliensis* and *A. flocculosa* completely lost virulence clusters, while another non-pathogenic member of the Ustilaginales, *Pseudozyma hubeiensis,* shows an almost complete set of effectors when compared to *U. maydis* (*Figure 3*).

**Table 2.** *MbA* proteins homologous to *U. maydis* effector genes with known virulence function.

| Name | Homologue | Query cover | E-value | Identity (%) | *U. maydis* knockout phenotype | Reference |
|---|---|---|---|---|---|---|
| g1653 | UMAG_01987 (Pep1) | 82% | 3-e56 | 60.96 | Complete loss of tumor formation – blocked in early stages of infection | *Doehlemann et al., 2009* |
| g1828 | UMAG_01829 (Afu1) | 99% | 0.0 | 71.57 | Organ-specific effector – reduced virulence in seedling leaves | *Schilling et al., 2014* |
| g2626 | UMAG_12197 (Cce1) | 98% | 1e-48 | 60.16 | Complete loss of tumor formation – blocked in early stages of infection | *Seitner et al., 2018* |
| g2765 | UMAG_11938 (Scp2) | 100% | 1e-73 | 93.44 | Reduced in virulence | *Krombach et al., 2018* |
| g2910 | UMAG_02475 (Stp1) | 32% | 3e-42 | 60.71 | Complete loss of tumor formation – blocked in early stages of infection | *Schipper, 2009* |
| g3652 | UMAG_02239 (See1) | 43% | 9e-11 | 54.90 | Organ-specific effector – reduced virulence in seedling leaves | *Redkar et al., 2015* |
| g3113 | UMAG_01375 (Pit2) | * | * | * | Complete loss of tumor formation – blocked in early stages of infection | *Doehlemann et al., 2011* |
| g3279 | UMAG_03274 (Rsp3) | 10% | 5e-20 | 70.11 | Strong attenuation of virulence – reduced tumor size and number | *Ma et al., 2018* |
| g5296 | UMAG_05731 (Cmu1) | 98% | 3e-70 | 43.84 | Reduced virulence | *Djamei et al., 2011* |
| g6183 | UMAG_06098 (Fly1) | 100% | 0.0 | 81.85 | Reduced virulence | *Ökmen et al., 2018* |
| g5835 | UMAG_05302 (Tin2) | 87% | 8e-24 | 37.81 | Minor impact on tumor formation – reduced anthocyanin biosynthesis | *Brefort et al., 2014* |

## Genetic characterization of *MbA*

To perform reverse genetics in *MbA*, we established a genetic transformation system based on protoplast preparation and polyethylene glycol (PEG)-mediated DNA transfer. In preliminary transformation assays, we expressed a cytosolic GFP reporter-gene under control of the constitutive *o2tef*-Promoter (*Figure 4A*). For the generation of knockout strains, a split marker approach was used to avoid ectopic integrations (*Figure 4B*). To allow generation of multiple knockouts, we used a selection marker-recycling system (pFLPexpC) that allows selection marker excision at each transformation round (*Khrunyk et al., 2010*).

We decided to apply the transformation system to study the *MbA* mating type loci in more detail. Although phylogenetically closely related to *U. hordei,* which has a bi-polar mating system, *MbA* owns a tetrapolar mating system whereby both mating type loci are physically not linked. This situation is similar to the mating type structure in the pathogenic smut *U. maydis* (*Figure 4A*). The *a*-locus, which encodes a pheromone receptor system that is required for sensing and fusion of compatible cells, is located on contig 6. The *b*-locus can be found on contig 1. This multiallelic mating locus contains two genes (*b-East* and *b-West*), which code for a pair of homeodomain transcription factors. Upon mating of compatible cells, pathogenic and sexual development are triggered by a heterodimeric bE/bW complex (*Brefort et al., 2009*). Since the *MbA* genome is completely equipped with mating type genes, we first deployed a screen for potential mating partners. To this end, we screened wild *M. bullatus* isolates to find a suitable mating partner, but we could not observe any mating event (*Figure 5—figure supplement 1*). To test if *MbA* is able to undergo pathogenic differentiation in the absence of mating, we generated a self-compatible strain (CB1) that carries compatible b-mating alleles: to construct the CB1 strain, we used compatible alleles of the *b-East* and *b-West* genes of the barley smut *U. hordei*, a pathogen that is the phylogenetically most closely related to *MbA* and amenable to reverse genetics. The native *MbA* locus was replaced by the compatible *U. hordei b-East* and *b-West* gene alleles via homologous recombination (*Figure 5B*).

Incubation of the *MbA* CB1 on charcoal plates led to the formation of aerial hyphae with the characteristic fluffy phenotype of filamentous strains like the self-compatible, solopathogenic *U. maydis* SG200 strain (*Figure 5C*). A second established method to induce filament formation in smuts is on

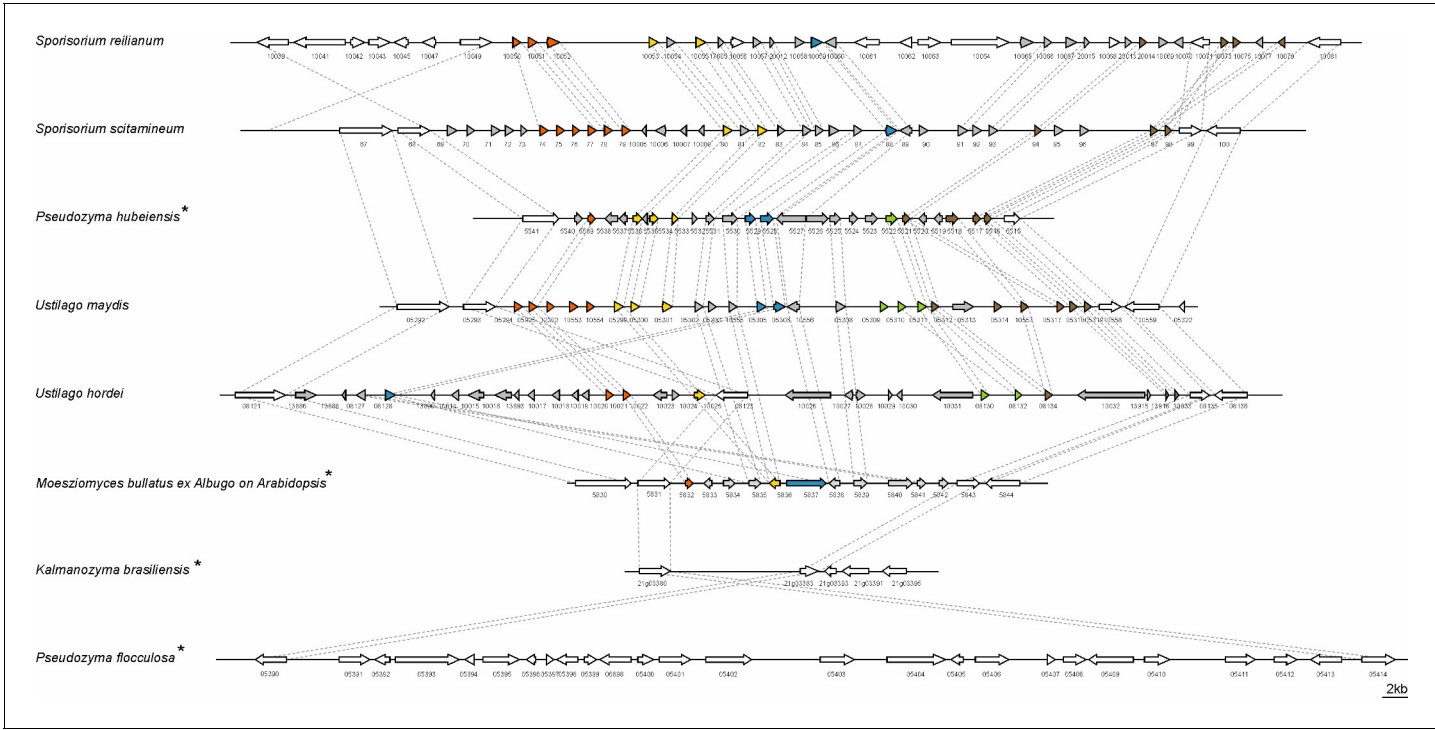

**Figure 3.** Structure of the largest virulence cluster (Cluster 19A) in pathogenic smut fungi and anamorphic smut yeasts (marked with*). Colors indicate genes with homology to each other: related gene families are indicated in orange, yellow, blue, green, and brown, whereas unique effector genes are shown in gray. Genes encoding proteins without a secretion signal are shown in white (*Brefort et al., 2014*).

The online version of this article includes the following figure supplement(s) for figure 3:

**Figure supplement 1.** Secondary metabolite gene in the genome of MbA.

**Figure supplement 2.** Protein alignment of the core effector Pep1 (*Hemetsberger et al., 2015*) from different Ustilaginomycetes.

**Figure supplement 3.** Comparison of known virulence clusters (*Kämper et al., 2006*) between *U. maydis* and *MbA*.

hydrophobic parafilm (*Mendoza-Mendoza et al., 2009*). Quantification after 18 hr incubation of *MbA* CB1 on parafilm resulted in the formation of filaments comparable to those of the *U. maydis* SG200 strain (*Figure 5D*). While about 17% of *MbA* wild-type cells showed filaments, the CB1 strain with compatible b-genes showed 38% filamentous growth.

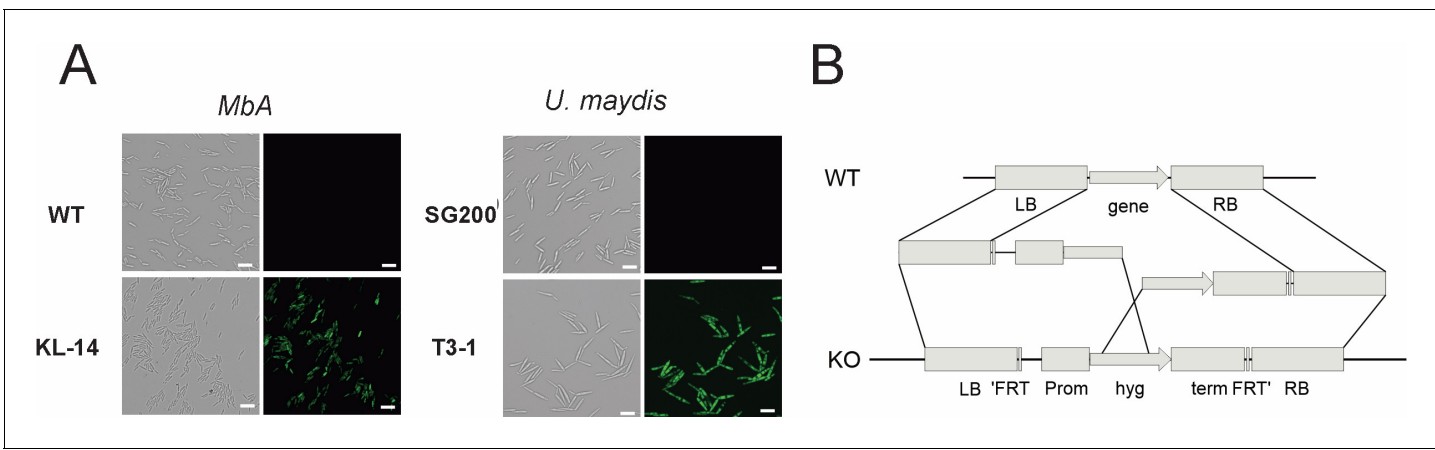

**Figure 4.** Genetic transformation of *MbA*. (A) Stable transformants that express cytosolic GFP could be obtained by generating protoplasts with Glucanex and ectopically integrating linear DNA fragments into the genome via polyethylene glycol (PEG)-mediated transformation. (B) Overview of the split-marker approach that was used to generate deletion mutants via homologous recombination.

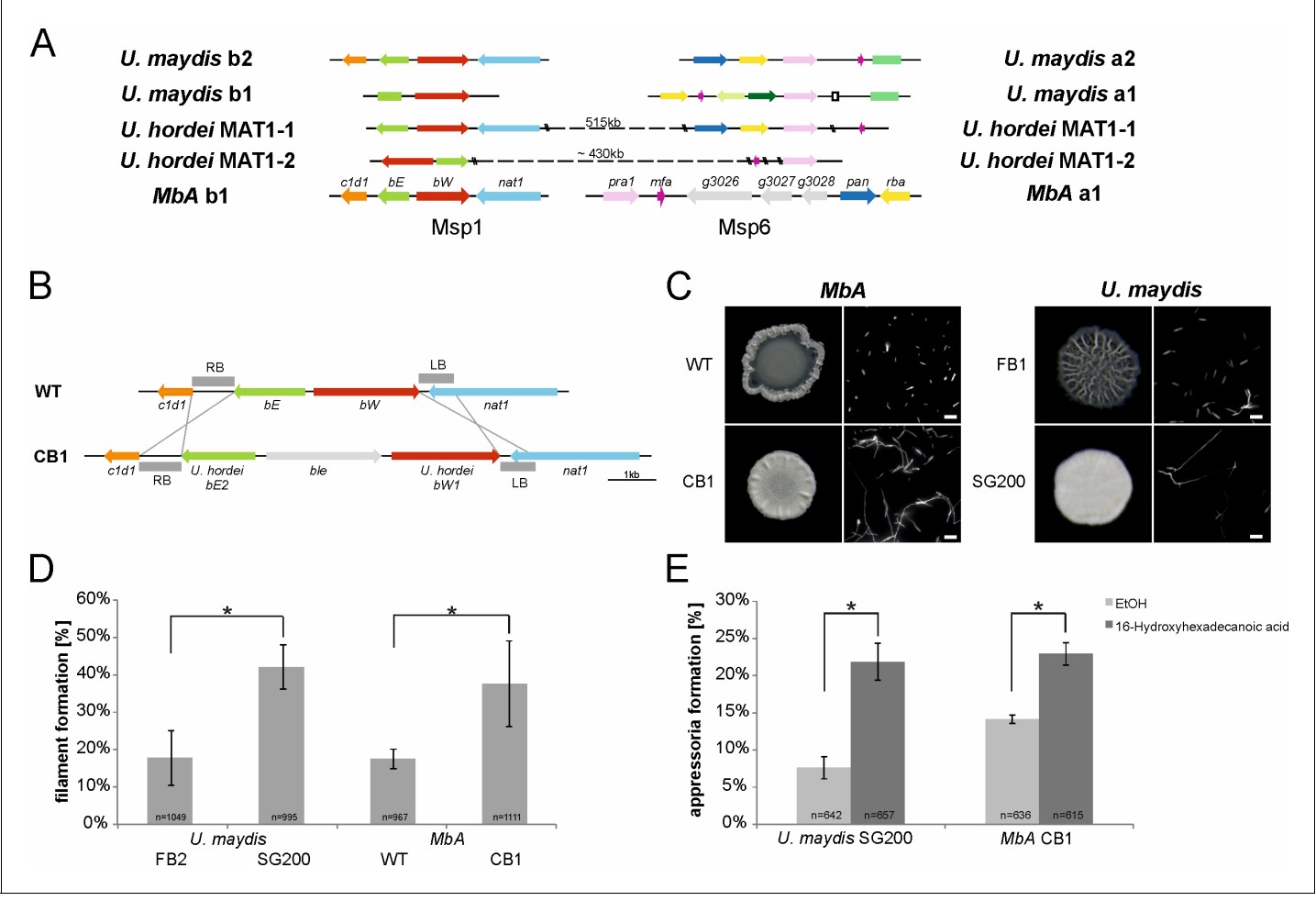

**Figure 5.** The self-compatible *MbA* strain CB1. (**A**) *MbA* mating type genes, unlike the ones of *U. hordei*, can be found on two different chromosomes similar to the tetrapolar mating type system of *U. maydis*. (**B**) To generate a self-compatible strain (CB1), the b-mating genes of *U. hordei* were integrated at the native *MbA* b-locus. (**C**) Unlike the *MbA* wild-type strain (top left), strain CB1 (bottom left) shows a fluffy phenotype on charcoal plates and filamentous growth. *U. maydis* haploid F1 strain (top right) and self-compatible SG200 strain (bottom right) were used as negative and positive control, respectively. (**D and E**) Induction of filamentation and appressoria formation in strain CB1 was studied in three independent experiments. For this around 1000 cells for filament formation and around 600 cells for appressoria formation were analyzed and error bars indicate standard error. After incubation on a hydrophobic surface, both, filament and appressoria formation in strain CB1, were significantly different (*chi-squared test for Independence – α = 0.0001) when compared to *MbA* wild type and similar to the level of the self-compatible *U. maydis* strain SG00. *U. maydis* haploid F2 strain was used as negative control (***Kämper et al., 2006***). Scale bar: 20 μm. 16-HDD: 16-hydroxyhexadecanoic acid.

The online version of this article includes the following source data and figure supplement(s) for figure 5:

**Source data 1.** Appressoria quantification of MbA.

**Figure supplement 1.** Mating assays of *MbA* and different *M. bullatus* isolates.

Formation of appressoria is a hallmark of pathogenic development in smut fungi (***Mendoza-Mendoza et al., 2009***). While the switch from yeast-like growth to filamentous development is the first step in the pathogenic development of smut fungi, host penetration is accompanied by the formation of a terminal swelling of infectious hyphae, termed 'appressoria'. Induction of appressoria formation in vitro can be induced by adding 100 μM of the cutin monomer 16-hydroxyhexadecanoic acid (HDD) to the fungal cells prior to cell spraying onto a hydrophobic surface (***Mendoza-Mendoza et al., 2009***). In the absence of HDD, only about 8% of the *U. maydis* SG200 cells and 14% of the *MbA* cells formed appressoria on parafilm 24 hr after spraying (***Figure 5E***). Addition of 100 μM HDD resulted in a significant induction of appressoria in both *U. maydis* and *MbA*, demonstrating that *MbA* does hold the genetic repertoire to form infection structures in vitro. Together, the analysis of the recombinant CB1 strain indicates that *MbA* can sense pathogenesis-related

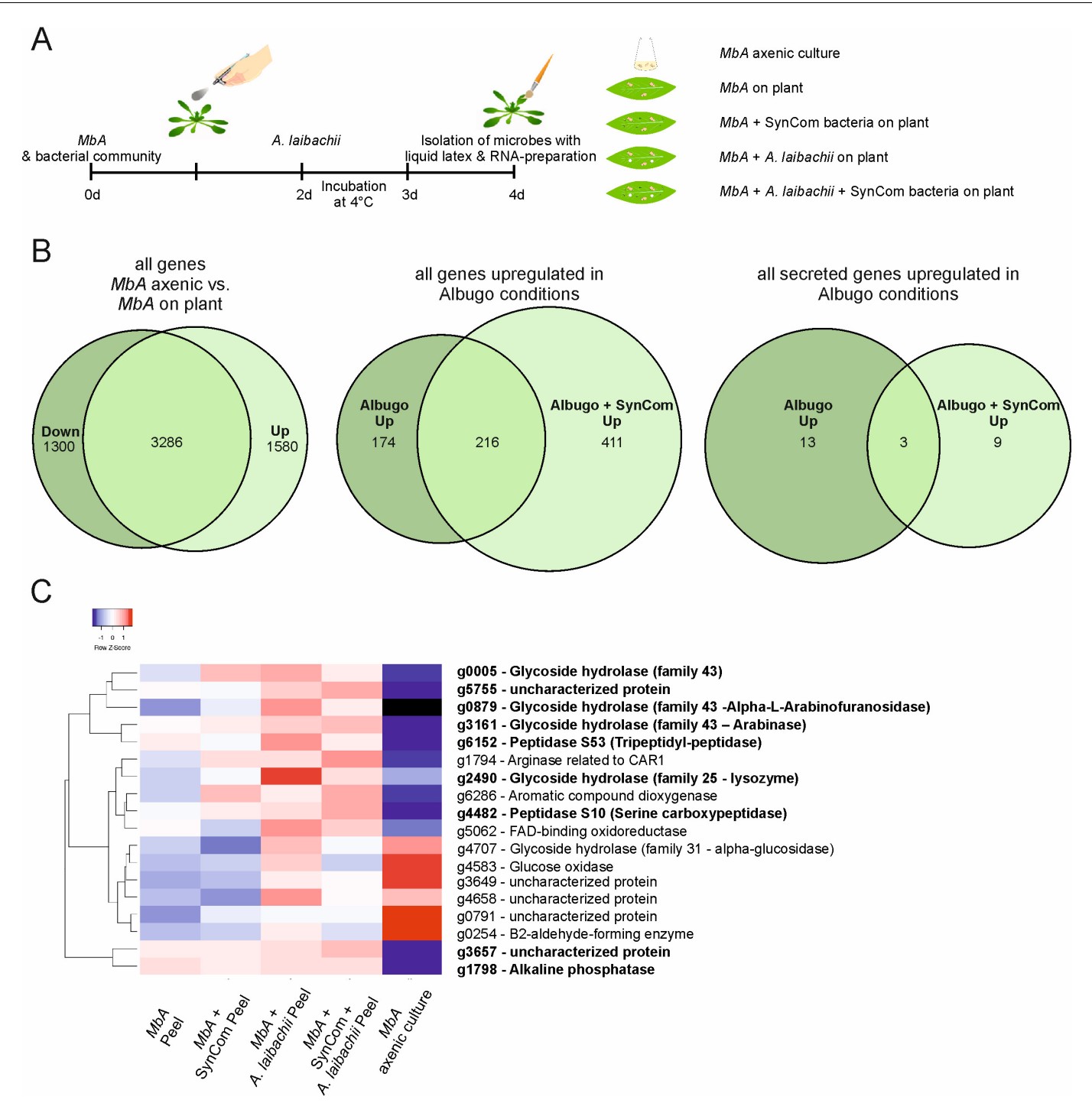

**Figure 6.** Transcriptome analysis of *MbA*. (**A**) Experimental setup used for the transcriptomic (RNA-Sequencing) analysis in *MbA*. (**B**) Venn diagrams showing differential regulated *MbA* genes after spraying of haploid cells onto the *A. thaliana* leaf surface. A total number of 801 genes were upregulated in response to *A. laibachii* in the presence and absence of bacterial SynCom. Of the 801 genes, 216 were upregulated in both conditions (*Supplementary file 2*). (**C**) Hierarchical clustering of the 18 *A. laibachii*-induced *MbA* genes that are predicted to encode secreted proteins. Of these genes, nine were selected as candidate microbe–microbe effector genes, based on their transcriptional upregulation and prediction to encode for extracellularly localized proteins.

The online version of this article includes the following source data and figure supplement(s) for figure 6:

**Source data 1.** Gene expression data of candidate effector genes.

**Figure supplement 1.** Gene expression analysis of MbA in response to SynCom bacteria and *A. laibachii*.

*Figure 6 continued on next page*

*Figure 6 continued*

**Figure supplement 2.** Gene ontology (GO) analysis of differentially regulated MbA genes.

surface cues and produce penetration structures to a similar level as that seen for the pathogenic model organism *U. maydis.*

## Identification of microbe–microbe effector genes by RNA-Seq

To study the transcriptomic response of *MbA* to different biotic interactions, RNA sequencing was performed. The *MbA* transcriptome was profiled in five different conditions (*Figure 6A*; cells in axenic culture versus cells on-planta, on-planta + SynCom, on-planta + *A. laibachii*, on-planta + SynCom + *A. laibachii*). Inoculations of *A. thaliana* leaves were performed as described above for *A. laibachii* infection assays (*Figure 6A*). For *MbA* RNA preparation, the epiphytic microbes were peeled from the plant tissue by using liquid latex (see Materials and methods section for details).

The libraries of the 15 samples (five conditions in three biological replicates each) were generated by using a poly-A enrichment and sequenced on an Illumina HiSeq4000 platform. The paired end reads were mapped to the *MbA* genome by using Tophat2 (*Kim et al., 2013*). The analysis revealed that *MbA* cells on *A. thaliana* leaves (on-planta) downregulated 1300 and upregulated 1580 genes compared to cells in axenic culture (*Figure 6B*, *Supplementary file 2*). Differentially expressed genes were determined with the 'limma'-package in R on 'voom' (*Figure 6—figure supplement 1*) using a False discovery rate threshold of 0.05 and log2FC > 0. A gene ontology (GO) terms analysis revealed that, among the downregulated genes, 50% were associated with primary metabolism (*Figure 6—figure supplement 2*). In the two conditions in which *A. laibachii* was present, we observed upregulation of 801 genes. Among these genes, 411 genes were specific to co-incubation of *MbA* with *A. laibachii* and SynCom while 174 were specific to incubation with *A. laibachii* only. A set of 216 genes was shared in both conditions (*Figure 6B*).

In the presence of *A. laibachii,* mainly metabolism- and translation-dependent genes were upregulated, which might indicate that *MbA* can access a new nutrient source in the presence of *A. laibachii* (*Figure 6—figure supplement 2*). Among all *A. laibachii*-induced *MbA* genes, 18 genes encode proteins carrying a secretion signal peptide and having no predicted transmembrane domain (*Figure 6C*). After excluding proteins being predicted to be located in intracellular organelles, nine candidate genes remained as potential microbe–microbe-dependent effectors, i.e. *MbA* genes that are induced by *A. laibachii*, show no or low expression in axenic culture, and encode for putative secreted proteins (*Figure 6C*). Interestingly, four of these genes encode putative glycoside hydrolases. Furthermore, two genes encode putative peptidases, one gene likely encodes an alkaline phosphatase and two encode uncharacterized proteins (*Figure 6C*).

To directly test the eventual antagonistic function of those genes toward *A. laibachii*, we selected the two predicted glycoside hydrolases-encoding genes *g5* and *g2490* (GH43 and GH25) and the gene encoding the uncharacterized protein *g5755* for gene deletion in *MbA*. The respective mutant strains were tested in stress assays to assess, whether the gene deletions resulted in general growth defects. Wild-type and mutant *MbA* strains were exposed to different stress conditions including osmotic stress (sorbitol, NaCl), cell wall stress (calcofluor, Congo-red), and oxidative stress ($H_2O_2$). Overall, in none of the tested conditions we observed a growth defect of the deletion mutants in comparison to wild-type *MbA* (*Figure 7—figure supplement 1*). To test an eventual impact of the deleted genes in the antagonism of the two microbes, the *MbA* deletion strains were each pre-inoculated on *A. thaliana* leaves prior to *A. laibachii* infection. Deletion of *g5* resulted in a significant but yet marginal increase of *A. laibachii* disease symptoms, while deletion of *g5755* had no effect on *A. laibachii*. We therefore considered these two genes being not important for the antagonism of *MbA* toward *A. laibachii*. Strikingly, the *MbA* Δ*g2490* strain almost completely lost its biocontrol activity toward *A. laibachii*. This phenotype was reproduced by two independent *g2490* deletion strains (*Figure 7A*). To check if this dramatic loss of microbial antagonism is specific to the deletion of *g2490*, *in-locus* genetic complementation of strain Δ*g2490_1* was performed via homologous recombination. The resulting strain *MbA* Δ*g2490/compl* regained the ability to suppress *A. laibachii*

infection, confirming that the observed phenotype specifically resulted from the deletion of the *g2490* gene (*Figure 7B*). Together, these results demonstrate that the biocontrol of the pathogenic oomycete *A. laibachii* by the basidiomycete yeast *MbA* is determined by the secretion of a previously uncharacterized GH25 enzyme, which is transcriptionally activated specifically when both microbes are co-colonizing the *A. thaliana* leaf surface.

## Functional characterization of the secreted MbA hydrolase

To characterize the protein function of the GH25 encoded by *MbA g2490*, we were using *Pichia pastoris* for heterologous expression. The recombinant protein was tagged with polyhistidine tag for Ni-NTA affinity purification. The purified protein was detected at an expected size of 27 kDa (*Figure 7—figure supplement 2*). In addition, via site directed mutagenesis a mutated version of the protein was generated, carrying a single amino exchange at the predicted active site (GH25_D124E). Both active and mutated versions of the GH25 hydrolase were subjected to a quantitative lysozyme activity assay using the fluorogenic substrate *Micrococcus lysodeikticus* with commercial Hen egg-white lysozyme (HEWL) as a control. We noticed a concentration-dependent increase in relative fluorescence unit (RFU)/min for the active GH25 in molar concentrations from 2 µM to 10 µM. Whereas, for similar concentrations, mutated GH25 (GH25mut) showed no significant increase in RFU/min compared to the active version. Commercial HEWL showed a steady increase in RFU/min from 1 µM to 5.5 µM concentrations (*Figure 7C*; *Figure 7—figure supplement 2*). Thus, the recombinant protein represents a functional GH25 hydrolase with a lysozyme activity.

To test for a direct function of the GH25 lysozyme, we treated *A. laibachii*-infected Arabidopsis plants with the recombinant protein. To quantify the impact of GH25 treatment on *A. laibachii* infection, we performed quantitative PCR to determine the relative *A. laibachii* biomass on Arabidopsis in response to GH25. Strikingly, we observed a significant reduction of *A. laibachii* colonization in leaves treated with the active GH25 lysozyme, while the mutated enzyme GH25_D124E did not significantly influence infection (p-value of <0.0001 and an $R^2$ value of 98.88%) (*Figure 7D*). Overall, treatment with the GH25 lysozyme reduced the amount of *A. laibachii* to about 50%.

## Discussion

Healthy plants in natural habitats are extensively colonized by microbes, therefore it has been hypothesized that the immune system and the microbiota may instruct each other beyond the simple co-evolutionary arms race between plants and pathogens (*Vannier et al., 2019*). Community members as individuals or in a community context have been reported to confer extended immune functions to their plant host. Root endophytic bacteria for example were found to protect *A. thaliana* and stabilize the microbial community by competing with filamentous eukaryotes (*Durán et al., 2018*). A large inhibitory interaction network was found in the leaf microbiome of *A. thaliana* and genome mining was used to identify over 1000 predicted natural product biosynthetic gene clusters (BGCs) (*Helfrich et al., 2018*). In addition, the bacterium *Brevibacillus* sp. leaf 182 isolate was found to inhibit half of the 200 strains isolated from *A. thaliana* phyllosphere. Further analysis revealed that *Brevibacillus* sp. leaf 182 produces a trans-acyltransferase polyketide synthase-derived antibiotic, macrobrevin along with other putative polyketide synthases (*Helfrich et al., 2018*).

In this study, we describe the role of the basidiomycete yeast *MbA*, which we previously co-isolated with the oomycete pathogen *A. laibachii* and now characterized as an antagonistic driver in the *A. thaliana* phyllosphere. *A. laibachii* inhibits in vitro growth of seven members of a bacterial leaf SynCom and, most strikingly, strongly suppresses disease progression and reproduction of the pathogenic oomycete *A. laibachii* on *A. thaliana*. *MbA* is a member of the Ustilaginales, which had previously been classified into the group of pathogenic smut fungi of the *Moesziomyces bullatus* species (*Kruse et al., 2017*). Our genome analysis identified the anamorphic yeasts *M. rugulosus*, *M. aphidis*, and *M. antarcticus*, which had previously been classified as 'Pseudozyma sp.', as the closest relatives of *MbA*. Anamorphic Ustilaginales yeasts are long known and have been used for biotechnological applications and also biocontrol (*Boekhout, 2011*). Mannosylerythritol lipids produced by *M. antarcticus* are known to act as biosurfactants and are of great interest for pharmaceutical applications (*Kitamoto et al., 1990*; *Morita et al., 2007*). Glycolipids like flocculosin produced by *A. flocculosa* or ustilagic acid characterized in the smut fungus *U. maydis* have antifungal activity.

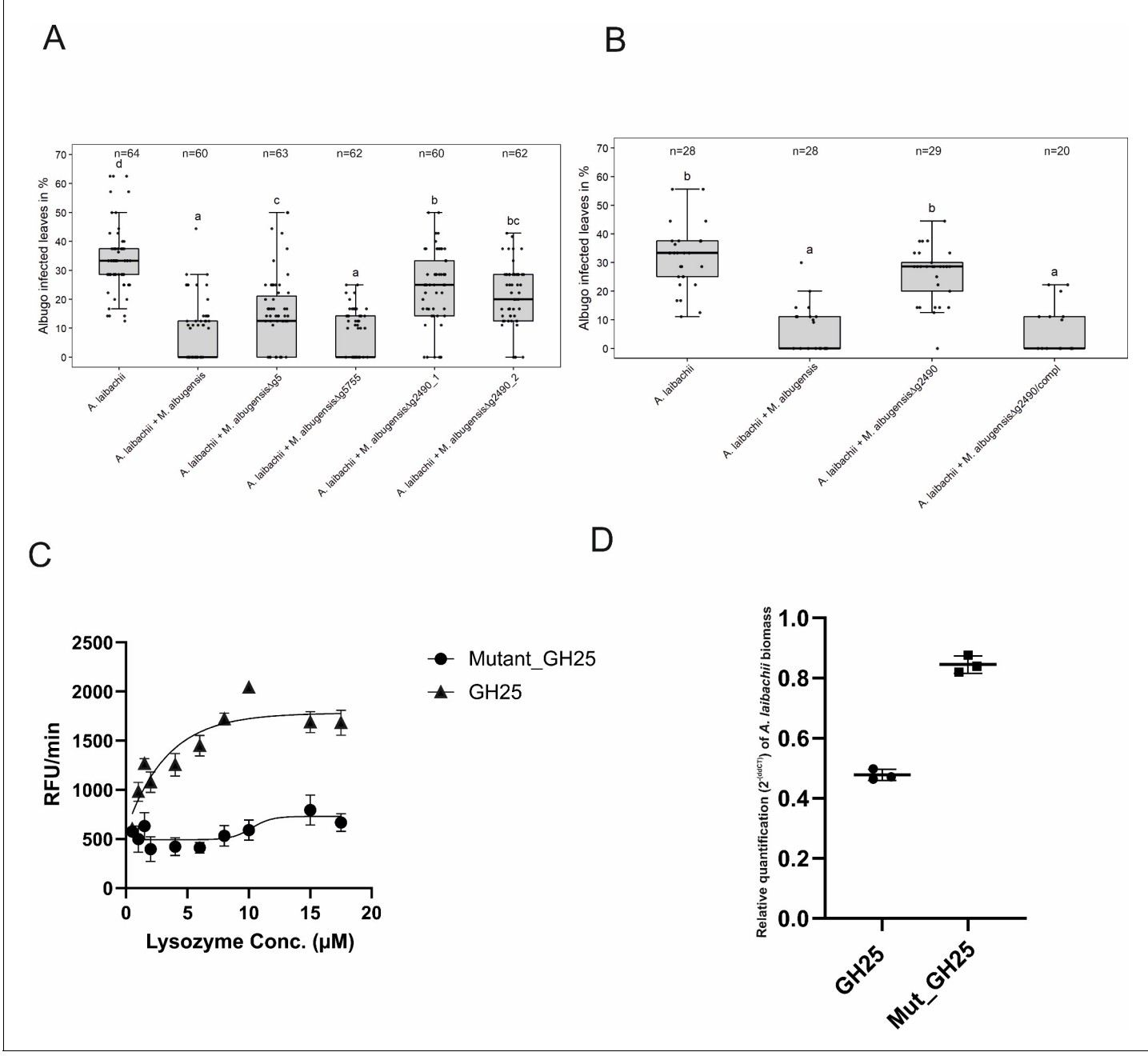

**Figure 7.** A reverse-genetic approach to identify the *MbA* gene that is responsible for the suppression of *A. laibachii* infection. (**A**) Three candidate microbe–microbe effector genes (*g5*, *g5755*, and *g2490*) were deleted in *MbA* and deletion strains were individually inoculated on *A. thaliana* together with *A. laibachii*. Inoculation of two independent *g2490 null* strains (Δ*g2490_1*; Δ*g2490_2*) resulted in significant and almost complete loss of the biocontrol activity of *MbA*. While deletion of g5 resulted in a marginal reduction of disease symptoms at 14 days post infection, deletion of g5755 had no effect on *A. laibachii*. (**B**) Genetic complementation of the *g2490* deletion restores the biocontrol activity to wild-type levels. Infections in (**A**) were performed in six, in (**B**) in three individual replicates. In each replicate 12 plants were infected. N indicates the number of infected plants that were scored for symptoms. Different letters indicate significant differences (p-values <0.05; ANOVA model for pairwise comparison with Tukey's HSD test). (**C**) Detection of lysozyme. Increasing concentrations of purified MbA_GH25 and MbA_GH25(D124E) were incubated with the DQ lysozyme substrate for an hour at 37°C. The fluorescence was recorded every minute in a fluorescence microplate reader using excitation/emission of 485/530 nm. Finally, relative fluorescence unit (RFU)/min was calculated for each concentration and plotted on the graph. Each data point represents three technical replicates and three independent biological replicates as indicated by the standard error measurement (SEM) bars. An unpaired t-test was performed for the active GH25 and Mutant_GH25 sets giving the p-value of <0.0001 and $R^2$ value of 77.24%. (**D**) Relative quantification of *A. laibachii* biomass in response to MbA_GH25 (active and mutant) treatment via qPCR. The oomycete internal transcribed spacer (ITS) 5.8 s was normalized to *A. thaliana* EF1-α gene to quantify the amount of *A. laibachii* DNA in the samples, 10 days post infection. Then relative biomass was calculated comparing control

*Figure 7 continued on next page*

*Figure 7 continued*

sets (only Albugo) with *A. laibachii* treated with GH25 and *A. laibachii* treated with Mutant_GH25 by ddCT method. Unpaired t-test between GH25 and Mutant_GH25 sets gave a p-value of <0.0001 and an $R^2$ value of 98.88%.

The online version of this article includes the following source data and figure supplement(s) for figure 7:

**Source data 1.** Infection data of *A. laibachii* on *A. thaliana*.

**Figure supplement 1.** Stress assay of *MbA* wild-type and knockout mutants of gene (g5, g5755, and g2490) respectively, on CM medium and 2% glucose (A) with different conditions (B: 100 µg/ml calcofluor; C: 150 µg/ml calcofluor; D: 1 mM $H_2O_2$; E: 45 µg/ml Congo-red; F: 1 M NaCl; G: 1 M sorbitol).

**Figure supplement 2.** Recombinant production of MbA_Gh25 for quantitative lysozyme assay.

**Figure supplement 3.** A molecular phylogenetic analysis using maximum likelihood estimation and based on pheromone receptor protein sequences similarity.

**Figure supplement 4.** Amino acid alignment of GH25 sequences from different fungi (see attached list 'GH25 with accession number', for full length sequences).

**Figure supplement 5.** Boxplot analysis of GH 25 treatment on in vitro *A. laibachii* zoosporangial germination in three biological replicates analyzing about 100 zoosporangial cells for each replicate.

Those compounds destabilize the membrane of different fungi and thus serve as biocontrol agents against powdery mildews or gray mold (*Cheng et al., 2003*; *Mimee et al., 2005*; *Teichmann et al., 2007*).

We identified 13 potential secondary metabolite gene clusters in *MbA*, including non-ribosomal peptide synthases and polyketide synthetase. Interaction among microbes within the same habitat is believed to have given rise to a variety of secondary metabolites (*Schroeckh et al., 2009*; *Rutledge and Challis, 2015*). The presence of *Streptomyces rapamycinicus* was shown to activate an otherwise silent polyketide synthase gene cluster, *fgnA*, in *Aspergillus fumigatus*. The resultant compound proved to be a potent fungal metabolite that inhibited the germination of *S. rapamycinicus* spores (*Stroe et al., 2020*). Therefore, secondary metabolite gene clusters and their corresponding products may confer a competitive advantage to fungi over the bacteria that reside in the same environment.

What is still under investigation is the relation of anamorphic yeasts with the related pathogenic smuts. Many smut fungi including the model species *U. maydis* are dimorphic organisms. In their saprophytic phase they grow as haploid non-pathogenic yeast cells. Only on appropriate host surfaces, haploid cells switch to filamentous growth and expression of pathogenicity-related genes is only activated upon mating in the filamentous dikaryon. A prime prerequisite for pathogenic development is therefore the ability of mating (*Bölker, 2001*; *Nadal and Gold, 2010*). Our genome analysis identified a tetrapolar mating system with a complete set of mating genes in *MbA*. Looking more closely on the phylogeny of different mating genes it appears that all sequenced Moesziomyces strains have the same pheromone receptor type (*Figure 7—figure supplement 3*). Together with our unsuccessful mating assays, this suggests that all sequenced strains of this species have the same mating type and, therefore, are unable to mate. Mating type bias after spore germination was reported for *Ustilago bromivora*, which leads to a haplo-lethal allele linked to the MAT-2 locus (*Rabe et al., 2016*). In this case, an intratetrad mating event rescues pathogenicity in nature as the second mating partner is not viable after spore germination. Together with the observation that anamorphic Moesziomyces yeasts are ubiquitous in nature, one could hypothesize that these fungi are highly competitive in their haploid form and antagonism might have led to the selection of one viable mating type. This eventually adapted to the epiphytic life style. At the same time, one should not dismiss the possibility that in the presence of a viable mating partner and suitable host surface, the yeasts can undergo potential sexual reproduction and thereby revitalizing the gene pool from time to time.

Transcriptome analysis showed that epiphytic growth of *MbA* on *A. thaliana* leads to massive transcriptional changes particularly in primary metabolism, which might reflect adaptation to the nutritional situation on the plant surface. Moreover, *MbA* showed specific transcriptional responses to a bacterial community, as well as to *A. laibachii* when being co-inoculated on plant leaves.

The presence of *A. laibachii* resulted in the induction of primary metabolism and biosynthesis pathways, which might reflect enhanced growth of *MbA* in the presence of *A. laibachii*.

A set of *MbA* genes encoding secreted hydrolases was induced by *A. laibachii* and one of these genes which encodes a putative GH25 hydrolase with similarity to Chalaropsis type lysozymes appeared to be essential for the biocontrol of *A. laibachii*. Initially discovered in the fungi *Chalaropsis* sp., this group of proteins is largely present in bacteria as well as phages, for example the germination-specific muramidase from *Clostridium perfringens* S40 (*Chen et al., 1997*). The bacterial muramidase, cellosyl from *Streptomyces coelicolor* (*Rau et al., 2001*), also belongs to the Chalaropsis type of lysozyme. These proteins are proposed to cleave the β-1,4-glycosidic bond between N-acetylmuramic acid (NAM) and N-acetylglucosamine (NAG) in the bacterial peptidoglycan. Specifically, the β-1,4 N,6-O-diacetylmuramidase activity allows the Chalaropsis type lysozyme to degrade the cell wall of *Staphylococcus aureus*, in contrast to the commercially available HEWL (*Rau et al., 2001*). Despite differences in structure and molecular weight from HEWL, the GH25 of MbA has lysozyme activity against the gram positive bacterium *Micrococcus lysodeikticus* in a fluorogenic assay. This highlights the overall biochemical functionality of the recombinant glycoside hydrolase. The glycoside Hydrolase 25 family is predicted to have an active site motif DXE that is highly conserved across the fungal kingdom (*Figure 7—figure supplement 4*). The structure of glycoside hydrolase family 25 from *Aspergillus fumigates* was characterized and the presence of N-terminal signal peptide was considered to indicate an extracellular secretion of the protein with possible antimicrobial properties (*Korczynska et al., 2010*). The role of the secreted hydrolase in the fungal kingdom is not completely explored yet. The presence of such hydrolases has in many cases been hypothesized to be associated with hyperparasitism of fungi parasitizing fungi (*Hyde et al., 2019*) or oomycetes parasitizing oomycetes (*Horner et al., 2012*). Our results might therefore indicate a cross kingdom hyperparasitism event between a fungus and an oomycete. Previous work on microbial communities has indicated that negative interactions stabilize microbial communities. Hyperparasitism is such a negative interaction with a strong eco-evolutionary effect on pathogen–host interactions and therefore on community stability (*Parratt and Laine, 2016*). *MbA* might therefore regulate *A. laibachii* infection and reduce disease severity. The qPCR evaluation of oomycete biomass strongly points toward the idea that *A. laibachii* is a direct target of antagonism for MBA. Since we observed reduced formation of *A. laibachii* in the presence of *MbA*, we also tested if the GH25 lysozyme would suppress zoospore germination. However, we could not detect a significant reduction of *A. laibachii* zoosporangia germination upon treatment with active GH25 lysozyme (*Figure 7—figure supplement 5*), suggesting that the GH25 lysozyme interferes with *A. laibachii* at a later stage of infection. As *A. laibachii* has been shown to reduce microbial diversity (*Agler et al., 2016*), *MbA* might increase diversity through hyperparasitism of *A. laibachii*. At the same time this increased diversity might have caused the need for more secondary metabolites to evolve in the *MbA* genome to defend against niche competitors. Through its close association with *A. laibachii*, *MbA* could be a key regulator of the *A. thaliana* microbial diversity and therefore relevant for plant health beyond the regulation of *A. laibachii* infection.

In conclusion, the secreted hydrolase we identified as a main factor of *A. laibachii* inhibition has great potential to act as antimicrobial agent. The isolated compound is not only valuable per se in an ecological context. It can further lay the grounds for exploring other microbial bioactive compounds that mediate inter-species and inter-kingdom crosstalk. A main goal of our future studies will be to understand on the mechanistic level, how the GH-25 suppresses *A. laibachii*, and at which developmental step the oomycete infection is blocked. It will be particularly interesting to elucidate if and how the GH25 enzyme activity directly interferes with the *A. laibachii* cell wall. While the canonical lysozyme substrate is found in bacterial cell walls, a detailed biochemical characterization of substrate specificity will be required to pinpoint potential target sites in the oomycete cell wall. Also, to the best of our knowledge there is no detailed information on the *A. laibachii* cell wall structure and composition. One could speculate if GH25 activity directly affects cell wall integrity of *A. laibachii*, or if a modification of the cell wall structure interferes with pathogenic development, e.g. by interfering with cellular differentiation, blocking signal perception, or by triggering a host defense response.

Since the GH-25 enzyme is well conserved among Ustilaginales including pathogenic species, it will also be tempting to elucidate whether the species-specific antagonism identified here is broadly conserved among Ustilaginales fungi and oomycetes. We further will investigate potential responses

by the host plant and how this impacts *A. laibachii* growth upon *MbA* colonization. Functional investigation of these interactions can provide meaningful insights as to why certain yeasts prefer to colonize specific environments. At the same time, it will be worth exploring how the basidiomycete yeasts influence the bacterial major colonizers of the phyllosphere.

# Materials and methods

## Key resources table

| Reagent type (species) or resource | Designation | Source or reference | Identifiers | Additional information |
|---|---|---|---|---|
| Gene (*MbA_g2490*) | g2490 | This paper | | |
| Strain, strain background (*Escherichia coli*) | DH5α | Other | | Doehlemann lab |
| Genetic reagent (*Pichia pastoris*) | KM71H-OCH | Other | | Doehlemann lab |
| Antibody | Monoclonal 6x-His tag antibody | Sigma (St. Louis; Mississippi; USA) | | 1/10,000 |
| Antibody | Mouse IgG (Monoclonal) | Thermo Fischer Scientific (Waltham; Massachusetts; USA) | | 1/3000 |
| Recombinant DNA reagent | pGAPzα (plasmid) | Invitrogen, Carlsbad, CA, USA | | |
| Sequence-based reagent | *A. thaliana* EF1-α:forward | *Ruhe et al., 2016* | PCR primers | AAGGAGGCTGCTGAGATGAA |
| Sequence-based reagent | *A. thaliana* EF1-α:reverse | *Ruhe et al., 2016* | PCR primers | TGGTGGTCTCGAACTTCCAG |
| Sequence-based reagent | Oomycete internal transcribed spacer (ITS) 5.8 s: forward | *Ruhe et al., 2016* | PCR primers | ACTTTCAGCAGTGGATGTCTA |
| Sequence-based reagent | Oomycete internal transcribed spacer (ITS) 5.8 s: reverse | *Ruhe et al., 2016* | PCR primers | GATGACTCACTGAATTCTGCA |
| Commercial assay or kit | EnzChek Lysozyme Assay Kit | Invitrogen | E22013 | Lysozyme activity assay |
| Chemical compound, drug | Trypan blue stain | Sigma Aldrich (No. 302643) | CAS-Number: 72-57-1 | |

## Strains and growth conditions

*MbA* wild-type strain was isolated from *A. laibachii* infected *A. thaliana* leaves [7]. Wild-type *MbA* (at 22°) and *U. maydis* (at 28°) strains were grown in liquid YEPS light medium and maintained on potato dextrose agar plates. King's B medium was used for culturing Syn Com bacterial members at 22°. All the strains were grown in a rotary shaker at 200 rpm. All the recipes for medium and solutions can be found in *Supplementary file 3*. Stress assays for fungi: wild-type and mutant strains of *MbA* grown to an optical density (600 nm) of 0.6–0.8 were centrifuged at 3500 rpm for 10 min and suspended in sterile water to reach an OD of 1.0. Next, a dilution series from $10^0$ to $10^{-4}$ was prepared in sterile $H_2O$. In the end, 5 µl of each dilution was spotted on CM plates supplemented with the indicated stress agents. The plates were incubated for 2 days at 22°C. Confrontation assays: at first,

MbA and SynCom bacterial strains were grown to an O.D of 0.8–1. MbA cultures (10 µl) were dropped in four quadrants of a potato dextrose agar plate, previously spread with a bacterial culture. Plates were incubated for 2–4 days at 22˚C.

## Transformation of *MbA* and plasmid construction for generation of knockout mutants

Fungal strains were grown in YEPSL at 22˚C in a rotary shaker at 200 rpm until an O.D. of 0.6 was reached and centrifuged for 15 min at 3500 rpm. The cells were washed in 20 ml of SCS (*Supplementary file 3*) and further centrifuged for 10 min at 3000 rpm, before being treated with 3 ml SCS solution with 20 mg/ml of Glucanex (Lysing Enzyme from *Trichoderma harzianum*, # L1412, Sigma). After 20 min of incubation at room temperature, as cell wall lysis occurred, cold SCS was added to the mixture and protoplasts spun down for 10 min at 2400 rpm. They were then washed twice with SCS and resuspended with 10 ml STC (*Supplementary file 3*) to be centrifuged at 2000 rpm for 10 min. Finally, the pellet was dissolved in 500 µl STC and stored in aliquots of 50 µl at −80˚ C. Five micrograms of plasmid DNA along with 15 µg heparin was added to 50 µl protoplasts. After incubation on ice for 10 min, STC/40% PEG (500 µl) was added to it and mixed gently by pipetting up and down; this step was followed by another 15 min on ice. The transformation mix was added to 10 ml of molten regeneration (reg) agar and poured over a layer of already solidified reg agar containing appropriate antibiotic solution. For the bottom layer, we used 400 µg/ml hygromycin/8 µg/ml carboxin/300 µg/ml nourseothricin (NAT).

Plasmids were cloned using *Escherichia coli* DH5α cells (Invitrogen, Karlsruhe, Germany). Construction of deletion mutants was performed by homologous recombination; the 5' and 3' flanking regions of the target genes were amplified and ligated to an antibiotic resistance cassette (*Kämper, 2004*). The ligated fragment was subsequently transformed into *MbA*. Homologous integration of the target gene was verified via PCR on the antibiotic resistant colonies. Oligonucleotide pairs for knockout generation and verification can be found in *Supplementary file 4*. PCR amplification was done using Phusion DNA polymerase (Thermo Scientific, Bonn, Germany), following the manufacturer's instructions, with 100 ng of genomic DNA or cDNA as template. Nucleic acids were purified from 1% TAE agarose gels using Macherey-Nagel NucleoSpin Gel and PCR Clean-up Kit.

## Mating assay and generation of the self-compatible *MbA* strain CB1

Haploid strains of *MbA* were grown in liquid cultures, mixed, and drops arranged on PD plates with charcoal to induce filament formation. Plate with the haploid *U. maydis* strains FB1 and FB2 and the solopathogenic strain SG200 served as internal control.

The complete b-locus of the solopathogenic *U. hordei* strain DS200 was amplified (*Figure 1—figure supplement 2*) and inserted into the *MbA* b-locus by homologous recombination. The strain obtained, known as compatible b1 (CB1), was tested positive by amplification of the right border and left border areas with primers specific for the genomic locus and for the plasmid region. Additionally, two primers specific for the *MbA bE* and *bW* genes were chosen to amplify parts of the native locus. To induce filament and appressoria formation in vitro we used a Moesziomyces YEPSL culture at $OD_{600}$ 0.6–0.8. The cells were diluted to an $OD_{600}$ of 0.2 in 2% YEPSL (for appressoria formation 100 µM 16-hydroxyhexadecanoic acid [Sigma-Aldrich] or 1% ethanol was added) and sprayed the yeast like cells on parafilm which mimics the hydrophobic plant surface. After 18 hr of incubation at 100% humidity the number of cells grown as filaments (or generating appressoria) was determined relative to the total number of total cells by using a light microscope.

## *Arabidopsis thaliana* leaf infections and quantification of albugo biomass quantification by qPCR

Sterilized *Arabidopsis thaliana* seeds were subjected to cold treatment for 7 days and sown on 1/2 strength Murashige Skoog (MS) medium (*Supplementary file 3*). The MS plates are directly transferred to growth chambers having 22˚C on a short-day period (8 hr light) with (33–40%) humidity and grown for 4 weeks before inoculation. Overnight liquid cultures of *MbA* and SynCom bacterial strains were grown to an $OD_{600}$ of 0.6. The cultures were spun down at 3500 rpm for 10 min and the pellets dissolved in $MgCl_2$. Five hundred microliters of each culture was evenly sprayed on 3-week old *A. thaliana* seedlings using airbrush guns. Two days later, a spore solution of *A. laibachii* was

then sprayed on the seedlings following the protocol of *Ruhe et al., 2016*. Two weeks later, the disease symptoms on the leaves were scored as a percentage between infected and non-infected leaves.

Four weeks old *A. thaliana* seedlings on MS plates were sprayed with *A. laibachii* as a control and GH25+ *A. laibachii* and *Mut_GH25+A. laibachii* as treatments. After 10 dpi, the seedlings were harvested, frozen in liquid nitrogen, and kept at −80°C. For DNA extraction, the frozen plant material was ground into a fine powder with mortar and pestle and treated with extraction buffer (50 mM Tris pH 8.0, 200 mM NaCl, 0.2 mM ethylenediaminetetraacetic acid [EDTA], 0.5% SDS, 0.1 mg/ml proteinase K [Sigma–Aldrich]). This was followed by centrifugation after the addition of one volume phenol/chloroform/isoamylalkohol, 25:24:1 (Roth). The top aqueous layer was removed and added to one volume of isopropanol to precipitate the nucleic acids. DNA pellet obtained after centrifugation was washed with 70% EtOH and finally dissolved in 50 µl nuclease-free water. For qPCR measurements, 10 µl of GoTaq qPCR 2× Master Mix (Promega, Waltham, Madison, USA), 5 µl of DNA (~50 ng), and 1 µl of forward and reverse primer (10 µM) up to a total volume 20 µl were used. Samples were measured in triplicates in a CFX Connect real-time PCR detection system (Bio-Rad) following the protocol of *Ruhe et al., 2016*. Amount of *A. laibachii* DNA was quantified using the following oligonucleotide sequences: *A. thaliana* EF1-α: 5′-AAGGAGGCTGCTGAGATGAA-3′, 5′-TGGTGGTCTCGAACTTCCAG-3′; Oomycete internal transcribed spacer (ITS) 5.8 s: 5′-ACTTTCAGCAGTGGATGTCTA-3′, 5′-GATGACTCACTGAATTCTGCA-3′. Cq values obtained in case of the oomycete DNA amplification was normalized to *A. thaliana* DNA amplicon and then the difference between control (only *Albugo*) and treatment (*Albugo*+ GH25/Mut_GH25) was calculated by ddCq. The relative biomass of *Albugo* was analyzed by the formula ($2^{-ddCq}$). Each data point in the graph represents three independent biological replicates.

## Nucleic acid methods

RNA-Extraction of Latex-peeled samples: Four weeks old *A. thaliana* plants were fixed between two fingers and liquid latex was applied to the leaf surface by using a small brush. The latex was dried using the cold air option of a hair dryer, carefully peeled off with a thin tweezer, and immediately frozen in liquid nitrogen. Afterwards, the frozen latex pieces were grinded with liquid nitrogen and the RNA was isolated by using Trizol Reagent (Invitrogen, Karlsruhe, Germany) according to the manufacturer's instructions. Turbo DNA-Free Kit (Ambion, Life Technologies, Carlsbad, California, USA) was used to remove any DNA contamination in the extracted RNA. Synthesis of cDNA was performed using First Strand cDNA Synthesis Kit (Thermo Fischer scientific, Waltham, Massachusetts, USA) according to recommended instruction starting with a concentration of 10 µg RNA. QIAprep Mini Plasmid Prep Kit (QIAGEN, Venlo, The Netherlands) was used for isolation of plasmid DNA from bacteria after the principle of alkaline lysis. Genomic DNA was isolated using phenol–chloroform extraction protocol (*Kämper et al., 2006*).

RT-qPCR oligonucleotide pairs were designed with Primer3 Plus. The oligonucleotide pairs were at first tested for efficiency using a dilution series of genomic DNA. The reaction was performed in a Bio-Rad iCycler system using the following conditions: 2 min at 95°C, followed by 45 cycles of 30 s at 95°C, 30 s at 61°C, and generation of melting curve between 65°C and 95°C.

## Bioinformatics and computational data analysis

Sequence assembly of *MbA* strains was performed using the HGAP pipeline (Pacific Biosciences). *MbA* genome was annotated with the Augustus software tool. Secretome was investigated using SignalP4.0. Analysis of functional domains in the secreted proteins was done by Inter-Pro Scan. Anti-Smash was used to predict potential secondary metabolite clusters. RNA sequencing was done at the Cologne Center for Genomics (CCG) by using a poly-A enrichment on an Illumina HiSeq4000 platform. The achieved paired end reads were mapped to the *MbA* and *A. thaliana* TAIR10 genome by using Tophat2 (*Kim et al., 2013*). RNA-Seq reads of *MbA* axenic cultures were used to generate exon and intron hints and to start a second annotation with Augustus. Heat maps were performed using the heatmap.2 function of the package gplots (version 3.0.1) in R-studio (R version 3.5.1). An analysis of variance (ANOVA) model was used for pairwise comparison of the conditions, with Tukey's HSD test to determine the significant differences among them (p-values <0.05).

## Heterologous protein production and GH25 activity assay

The *Pichia pastoris* KM71H-OCH gene expression system was used to produce MBA_GH25 domain tagged with an N-terminal Polyhistidine tag (6xHis) and a C-terminal peptide containing the c-myc epitope and a 6xHis tag. The His-MspGH25 cloned into pGAPZαA vector (Invitrogen, Carlsbad, CA, USA) under the control of a constitutive promotor with an α-factor signal peptide for secretion. Expression and purification of recombinant proteins were performed according to manufacturer's instructions (Invitrogen Corporation, Catalog no. K1710-01): YPD medium supplemented with 100 µg ml$^{-1}$ zeocin was used for initial growth of *P. pastoris* strains at 28°C and 200 rpm (for liquid cultures). Production of the recombinant protein was performed in 1 L buffered (100 mM potassium phosphate buffer, pH 6.0) YPD medium with 2% sucrose at 28°C for 24 hr with 200 rpm shaking. Next the protein was subjected to affinity purification with a Ni-NTA-matrix according to manufacturer's instructions (Ni-Sepharose 6 Fast-Flow, GE-Healthcare; Freiburg, Germany). After purification, the His-MspGH25 protein was dialyzed in an exchange buffer (0.1 M NaPi, 0.1 M Nacl, pH = 7.5). The purified protein was kept in 100 µl aliquots at 4°C.

Site-directed mutagenesis was performed on pGAPZα-His- MspGH25 vector according to the instructions of the QuikChange Multi Site-Directed Mutagenesis Kit (Agilent Technologies, Santa Clara, United States) with primers targeting nucleotides of the active site of GH25.

Purified glycoside hydrolase of *MBA* from *P. pastoris* was quantified according to a sensitive fluorescence-based method using Molecular Probes EnzChek Lysozym-Assay-Kit (ThermoFisher Scientific, Katalognummer: E22013). DQ lysozyme substrate (*Micrococcus lysodeikticus*) stock suspension (1.0 mg/ml) and 1000 units/ml HEWL stock solution were prepared according to the manufacturer's instruction. Molar concentration of the HEWL stock solution was calculated using the following website (https://www.bioline.com/media/calculator/01_04.html) and was found to be 11 µM. Protein concentration of MspGH25 both active and mutated version was measured in the Nanodrop 2000c spectrophotometer (Thermo Fischer scientific, Waltham, Massachusetts, USA) according to the manufacturer's instructions using 100 µl of sample after using 100 µl of the appropriate buffer as a blank control in glass cuvette. The molar concentrations of recombinant proteins were also calculated as above.

At the start of the reaction 50 µl of the DQ lysozyme substrate working suspension was added to each microplate well containing reaction buffer with either HEWL (in molar concentrations ranging from 0.1 to 5.5 µM) or MspGH25 (in molar concentration from 0.5 to 17.5 µM). Fluorescence intensity of each reaction was measured every 5 min to follow the kinetic of the reaction at 37°C for 60 min, using fluorescence microplate reader with fluorescein filter Tecan Infinite 200 Pro plate reader (Tecan Group Ltd., Männendorf, Switzerland). Digestion products from the DQ lysozyme substrate have an absorption maximum at ~494 nm and a fluorescence emission maximum at ~518 nm.

## Acknowledgements

This work was funded through by the Deutsche Forschungsgemeinschaft (DFG, German Research Foundation) under Germany's Excellence Strategy EXC-2048/1, Project ID 390686111, and the DFG priority program SPP2125 'DECRyPT'. We are grateful to Marco Thines for generously providing *M. bullatus* wild-type strains. We thank Libera Lo Presti for critically reading the manuscript and helpful comments and suggestion.

## Additional information

### Funding

| Funder | Grant reference number | Author |
| --- | --- | --- |
| Deutsche Forschungsgemeinschaft | SPP 2125 DECRyPT | Katharina Eitzen Priyamedha Sengupta |
| Deutsche Forschungsgemeinschaft | EXC-2048/1 | Katharina Eitzen |
| Deutsche Forschungsgemeinschaft | Project ID 390686111 | Katharina Eitzen |

The funders had no role in study design, data collection and interpretation, or the decision to submit the work for publication.

### Author contributions

Katharina Eitzen, Conceptualization, Investigation, Visualization, Methodology, Writing - original draft; Priyamedha Sengupta, Investigation, Visualization, Methodology, Writing - review and editing; Samuel Kroll, Investigation, Methodology; Eric Kemen, Gunther Doehlemann, Conceptualization, Formal analysis, Supervision, Funding acquisition, Writing - original draft, Project administration, Writing - review and editing

### Author ORCIDs

Gunther Doehlemann (iD) https://orcid.org/0000-0002-7353-8456

### Decision letter and Author response

Decision letter https://doi.org/10.7554/eLife.65306.sa1
Author response https://doi.org/10.7554/eLife.65306.sa2

## Additional files

### Supplementary files

• Supplementary file 1. Composition of the bacterial SynCom.

• Supplementary file 2. *MbA* gene expression data.

• Supplementary file 3. Growth media and buffers used in this study.

• Supplementary file 4. PCR primers used in this study.

• Transparent reporting form

### Data availability

Genome information and RNA sequencing have been submitted to NCBI Genbank and are available under the following links: https://www.ncbi.nlm.nih.gov/geo/query/acc.cgi?acc=GSE148670.

The following dataset was generated:

| Author(s) | Year | Dataset title | Dataset URL | Database and Identifier |
|---|---|---|---|---|
| Eitzen K, Doehlemann G, Kemen E | 2021 | Transcriptome of Moesziomyces albugensis in response to different biotic factors (A. laibachii & SynCom) on A. thaliana leaves | https://www.ncbi.nlm.nih.gov/geo/query/acc.cgi?acc=GSE148670 | NCBI Gene Expression Omnibus, GSE148670 |

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
