## [Decision Letter]

**Acceptance summary:**

Understanding functions and mechanisms of microbe-microbe interactions in plant microbiota communities is of great fundamental interest, and promising for agricultural application for example in biocontrol. This study reports on a specific molecular player mediating antagonism by a leaf-surface dwelling yeast on an oblicate biotrophic oomycete pathogen. The yeast *Moesziomyces bullatus ex Albugo* (MbA) was isolated from the biotrophic pathogen oomycete *Albugo laibachii*, which acts as a hub organism in Arabidopsis phyllosphere microbiome. The study promotes the establishment of Mba as a leaf epiphytic model fungus, by providing a sequenced genome, the transcriptome in response to interaction with a phyllosphere bacterial SynCom and *A. laibachii* and by establishing a transformation system for generation of k. o. strains by homologous recombination. The authors report that MbA antagonizes the oomycete pathogen *A. laibachii*, via a putatively secreted GH25 hydrolase with lysozyme activity. This important finding provides novel insights into the yet elusive molecular mechanisms involved in shaping the microbiome community on leaf surfaces through microbe-microbe interactions.

**Decision letter after peer review:**

[Editors’ note: the authors submitted for reconsideration following the decision after peer review. What follows is the decision letter after the first round of review.]

Thank you for submitting your work entitled "A fungal member of the *Arabidopsis thaliana* phyllosphere antagonizes *Albugo laibachii* via a putative secreted hydrolase" for consideration by *eLife*. Your article has been reviewed by three peer reviewers, and the evaluation has been overseen by a Reviewing Editor and a Senior Editor, and three reviewers. The reviewers have opted to remain anonymous.

Our decision has been reached after consultation between the reviewers. Based on these discussions and the individual reviews below, we regret to inform you that your work will not be considered further for publication in *eLife* in its present form.

Although the reviewers and ourself found the study original and the establishment of *Moesziomyces albugensis* as a genetic model as well as its characterization interesting, the reviewers detected several flaws, which prevent publication in *eLife*:

– the phylogenetic placing of "*Moesziomyces albugensis*" appears incorrect

– its mode of action on Albugo is not described and

– the GH25 enzyme and its function are not sufficiently characterized

We would be happy to consider an improved version of the manuscript, in which the reviewer comments have been addressed comprehensively; and which provides a better understanding of *Moesziomyces* action on Albugo and GH25 function.

Reviewer #1:

The authors establish *Moesziomyces albugensis* as a new "model fungus" at the leaf surface, which may act as a parasite on the pathogenic oomycete of Arabidopsis *Albugo laibachii*. They provide a sequenced genome, the transcriptome in response to interaction with the plant, phyllosphere bacteria and the oomycete *A. laibachii*, a transformation system for generation of k. o. strains by homologous recombination and characterize the mating type as well as filament and appressoria forming behavior in vitro.

In addition, they also report some interesting biology and show that *M. albugensis* antagonizes the oomycete pathogen *A. laibachii*, probably via a secreted putative GH25 hydrolase. This finding and other findings, that may be expected from this research line in the future, may yield biocontrol fungi to combat agriculturally relevant oomycete infections of crop leaves.

While M.albugensis is an exciting new fungus and the identification of the effector required for parasitism of *A. laibachii* is very interesting, the manuscript still suffers from a few unclarities.

– From the text I did not understand whether M. albugensis lives only epiphytically or can also colonize the plant tissue (it forms appressoria on hydrophobic surfaces) or whether it is exclusively associated with *A. laibachii*. This should be clarified.

– It is unclear how the bacterial SynCom was constructed and which isolates it contains.

– The quantification of *A. laibachii* infection seems kind of rough and prone to subjectivity. An additional, more quantitative method should be used for quantification, for example by qPCR using an amplicon from *A. laibachii* DNA normalized to a plant amplicon.

– It is unclear whether GH25 hydrolase suppresses infection of plant leaves by *A. laibachii* or whether it helps *M. albugensis* to parasitize *A. laibachii* by other means. The authors should examine the effect of isolated GH25 hydrolase on *A. laibachii* infection.

– At which stage of infection is *A. laibachii* blocked by GH25 hydrolase or *M. albugensis*?

Reviewer #2:

The study by Eitzen et al., characterizes a yeast species isolated from *Arabidopsis* phyllosphere classified as *Moesziomyces albugensis* (Ma). The report provides phylogenetic classification, genome analysis and functional analysis of Ma inhibitory activity towards the white rust oomycete *Albugo laibachii* (Al). Understanding functional interaction in the plant microbiota is a field of intense research with a great fundamental interest, as well as promising applied perspectives regarding the identification of novel biocontrol agents such as potentially Ma. I found the manuscript very pleasant to read and well structured. The work appeared rigorous and conclusions are appropriately supported by the available data. A number of biocontrol yeasts have been isolated in the past, but in my opinion this study stands out by providing direct genetic evidence of the anti-rust activity of Ma, through the identification of one underlying gene (encoding a putative GH25 enzyme) and the characterization of the corresponding mutant phenotype. The Ma genome sequence and gene expression data also provide an interesting resource for further analysis of the biology of this phyllosphere fungus.

1) I would have liked to see the activity of the GH25 characterized further. I am lacking a clear hint towards GH25 molecular activity on Al and possibly other members of the phyllosphere. Does ectopic (eg in plants) expression of GH25 has inhibitory effect on Albugo? Is there any indication on which species would this enzyme act? Is it acting specifically on Albugo? What protects Ma from its own secreted hydrolase? How conserved in GH25 across fungi? Is there anything specific about its sequence (and predicted activity) as compared to other GH25s? to other glycosyde hydrolases? To what extent is GH25 activity beneficial to Ma? Could the growth of this fungus measured on Albugo-infected vs Albugo-free plants? in vitro with or without Albugo?

I believe some info and discussion on these questions is important as the identification of GH25 is in my view the major novelty of this work.

2) The description of statistical analyses needs attention. I particular, I could not find information on how gene expression and differentially expressed genes were determined (see statistical comments).

Reviewer #3:

The manuscript deals with the characterization of a phyllosphere yeast that is supposed to have an antagonistic effect on *Albugo laibachii*.

While the manuscript is well written, it suffers from numerous cases of overinterpretation of the results and methodological flaws, as outlined below.

Abstract. It is beyond reason that the authors call their yeast isolate "*Moesziomyces albugensis*". First, there is not even a remotely valid description of the "species", and second, it is quite obvious that after checking some of the recent literature on *Moesziomyces* (part of which two of the authors even co-authored) that the isolate they are investigating is clearly *Moeziomycesbullatus*. This needs to be changed throughout the manuscript.

Abstract. That the isolates "prevents" Albugo infections is insufficiently shown (but see later criticism on this point).

Abstract. The evidence for the claim that GH25 is a major effector of the antagonism is weak, at best.

Introduction. This statement is not correct. there are several studies, e.g. by the Macia-Vicente group that have shown that the local environment drives the diversity of root-associated microbiomes.

Introduction. The genus Pseudozyma is obsolete. This should be termed Moesziomyces. And the species is known as well, it is *Moesziomyces bullatus*.

Introduction. I was surprised about the statement and have the read Kurse et al., manuscript. In fact, the authors report the opposite, an infrequent sexual reproduction on grass hosts as the most probable explanation for the retainment of a sexual reproduction.

Introduction. I really wonder why the authors cite this less comprehensive phylogeny, while two of them have shown, based on a broader sampling that in fact the species mentioned are embedded within *Moesziomyces* and that *Pseudozyma aphidis* is conspecific with *Moesziomyces bullatus*.

Introduction. The transfer of these species into *Pseudozyma* had already been done, thus, "can be renamed" is not a correct statement.

Introduction. This should read *Moesziomyces bullatus* or "A strain formerly classified as *Pseudozyma aphidis* (now *Moesziomyces bullatus*)".

Introduction. There is no such species name as *Moesziomyces albuginensis*. And this also would not make phylogenetic or evolutionary sense, as the *Moesziomyces bullatus* strain isolated from *Arabidopsis thaliana* is firmly embedded within *Moesziomyces bullatus* and there is no evidence that is should be considered as a biologically distinct species. Also, the evidence of disease "prevention" are at most anecdotal (see later criticism).

Results. This statement alone refutes the previous statement in the Introduction. If the species was isolated from and found to be closely associated with the spores of Albugo, even over years of continuous cultivation cycles, how can the authors confidently claim that the species prevents infection?

Introduction. There are many strains of *Moesziomyces bullatus* (as *Pseudozymaaphidis*) that are marked as biocontrol agent. The new strain would just be an addition to that, there is little novelty in this.

Results. This argumentation is impossible to follow. Just because something is isolated from a special location does not make it a new species.

Results. I really cannot see the reason for the authors claiming again the identification was just "*Pseudozyma sp*." while they could show in a previous manuscript [Kruse et al., 2017] that the species is Meosziomyces bullatus.

Results. This interaction does not need to be specific. The test against *Ustilago maydis* is also difficult to justify, as that species is largely unrelated to *Moesziomyces bullatus*. Instead, other species of the genus *Moesziomyces*, especially additional strains of *Moesziomyces bullatus* should have been used.

Results. While in general, it is interesting that the yeast showed an antogonistic effect, dose dependency should have been checked. Several species of the Ustulagoinales are known to produce biosurfactants that alter the properties of the plant cuticle. Thus, the effect observed can probably already explained by this. This could be addressed by additional experiments with a variety of Ustilago and *Moesziomyces* strains and demonstrating that a lower dose of the Albugo-associated strain inhibits the infection.

Subsection “The genome of *Moesziomyces albugensis”*. This lacks a lot of the usual quality control, such as genome completeness statistics, read mapping specifics, testing of the gene predictions by RNA mapping.

Subsection “The genome of *Moesziomyces albugensis”*. Evidence should be presented that this is not the result of a mis-assembly.

Subsection “The genome of *Moesziomyces albugensis”*. Even though "tempting", this speculation of the authors seems very far-fetched, considering that there are many unrelated groups of the Ustilaginales that cause tumors or not.

Subsection “The genome of *Moesziomyces albugensis”*. This part is interesting, but needs to be backed up by RNASeq and metabolic profiling.

Subsection “The genome of *Moesziomyces albugensis”*. If the authors reanalyzed that genome, they could probably see that most of the "effector losses" are the result of bad gene calls.

Subsection “The genome of *Moesziomyces albugensis”*. I wonder why the authors do not cite or discuss their own findings on several *Moesziomyces* species previously characterized as non-pathogenic. Ökmen et al., had found years ago that these yeasts possess functional copies of PEP1.

Subsection “The genome of *Moesziomyces albugensis”*. The interpretation here cannot be upheld. Given current data, it seems to be a specialty of Ustilago madis that the clusters are expanded (see e.g. Sharma et al., 2014). In addition, the "loss" or presence of the clusters is likely a simple function of phylogenetic distance. Judging from the unfortunately not well resolved phylogeny in [Wang et al., 2015], Kalmanozyma could simply be most unrelated to *Ustilago maydis*.

Subsection “Genetic characterization of *M. albugensis”*. This is the first part of the manuscript that seems solid, even though I wonder why non infection trials on the natural host of *Moesziomyces bullatus* were done.

Subsection “Identification of microbe-microbe effector genes by RNA-Seq”. While this cell biology part is again rather solid, the conclusions seem much too strong. For stating that the secretion of GH25 is important, the authors would need to construct a non-secreted version under a native promotor. In addition, the metabolic profile would need to be analysed, both of the secreted metabolites and those retained in the cells. It is conceivable that the GH25 enzyme processes a non-effective precursor to generate the observed effect.

As many results obtained are already discussed in the Results section, the Discussion section is sometimes repetitive. I encourage the authors to keep the Results section free from interpretation and to do this in the Discussion section. As many incorrect claims are made again in the Discussion, I do not comment on them again.

Subsection “Bioinformatics and computational data analysis”. The information regarding the genome assembly and gene prediction is at best minimal. This needs to much extended, also including the necessary quality checks as indicated earlier.

Figure 1. This is hardly useful to present the "phylogenetic position" of the strain. There are much more elaborate phylogenies and the authors should simply refer to them. Also, the separation into "apathogenic" and "pathogenic" strains is pointless, as they confirm what was previously known. Likely there are no apathogenic members of the Ustilaginales and the yeasts have just by chance been isolated first from the environment before the pathogenic stage was discovered.

[Editors’ note: further revisions were suggested prior to acceptance, as described below.]

Thank you for submitting your article "A fungal member of the *Arabidopsis thaliana* phyllosphere antagonizes *Albugo laibachii* via a secreted lysozyme" for consideration by *eLife*. Your article has been reviewed by three peer reviewers, including Caroline Gutjahr as the Reviewing Editor and Reviewer #1, and the evaluation has been overseen by a Reviewing Editor and Christian Hardtke as the Senior Editor.

The reviewers have discussed the reviews with one another and the Reviewing Editor has drafted this decision to help you prepare a revised submission.

We would like to draw your attention to changes in our revision policy that we have made in response to COVID-19 (https://elifesciences.org/articles/57162). Specifically, we are asking editors to accept without delay manuscripts, like yours, that they judge can stand as *eLife* papers without additional data, even if they feel that they would make the manuscript stronger. Thus the revisions requested below only address clarity and presentation, except if you prefer yourself to add data regarding request No. 3.

Summary:

Eitzen et al., investigate a specific mechanism of antagonism by an epiphytic yeast on an obligate biotrophic oomycete pathogen. They established the yeast *Moesziomyces bullatus ex Albugo* (MbA), which represents a hub organism in the Arabidopsis phyllosphere microbiome, as a leaf epiphytic model fungus, by providing a sequenced genome, the transcriptome in response to interaction with a phyllosphere bacterial SynCom and the oomycete *A. laibachii* and by establishing a transformation system for generation of k. o. strains by homologous recombination.

They report that M. bullatus ex Albugo antagonizes the oomycete pathogen *A. laibachii*, via a putatively secreted GH25 hydrolase with lysozyme activity. This finding, and other findings that may be expected from this research line in the future, do not only add to a molecular understanding of microbial interactions on the leaf surface but may in the future yield biocontrol fungi to combat agriculturally relevant oomycete infections of crop leaves.

Essential revisions:

Based on the major comments of all three reviewers and a discussion, we request to address the following points before the manuscript can be accepted by *eLife*:

1) The reviewers found the phylogenetic tree in Figure 1 misleading (see also comments about the tree in the previous version of the manuscript). Since a more comprehensive and correct tree is published in Kruse et al., Figure 1 should be removed from the manuscript.

2) The GH25 hydrolase exhibits lysozyme activity, which hydrolyzes β‐(1,4) linkages between the NAM and NAG saccharides. These glucans are usually found on bacterial cell walls. However, to the reviewer's knowledge, the cell wall of oomycetes is composed of β-1,3, and β-1,6 glucans. How does GH25 hydrolase inhibit the oomycete *Albugo laibachii*? This question or hypotheses should be raised and discussed in the Discussion section.

3) The Title states that GH25 is a secreted enzyme. Unless the authors include an experiment showing secretion, statements on secretion should be toned down in the title and across the manuscript where applicable. (In several instances the authors have already carefully written about "an enzyme with predicted secretion peptide").

---

## [Author Response]

[Editors’ note: the authors resubmitted a revised version of the paper for consideration. What follows is the authors’ response to the first round of review.]

Reviewer #1:The authors establish *Moesziomyces albugensis* as a new "model fungus" at the leaf surface, which may act as a parasite on the pathogenic oomycete of Arabidopsis *Albugo laibachii*. They provide a sequenced genome, the transcriptome in response to interaction with the plant, phyllosphere bacteria and the oomycete *A. laibachii*, a transformation system for generation of k. o. strains by homologous recombination and characterize the mating type as well as filament and appressoria forming behavior in vitro.In addition, they also report some interesting biology and show that *M. albugensis* antagonizes the oomycete pathogen *A. laibachii*, probably via a secreted putative GH25 hydrolase. This finding and other findings, that may be expected from this research line in the future, may yield biocontrol fungi to combat agriculturally relevant oomycete infections of crop leaves.While M.albugensis is an exciting new fungus and the identification of the effector required for parasitism of *A. laibachii* is very interesting, the manuscript still suffers from a few unclarities.– From the text I did not understand whether *M. albugensis* lives only epiphytically or can also colonize the plant tissue (it forms appressoria on hydrophobic surfaces) or whether it is exclusively associated with *A. laibachii*. This should be clarified.

We apologize for being unclear in this point. *M. albugensis* belongs to a group of basidiomycete yeasts that has been found to lead epiphytic existence on the leaf surface. However, since the recombinant, self-compatible strain (CB1) which we generated in this study was found to form appressoria on hydrophobic surface, this hints towards the idea that in nature, the anamorphic epiphytic yeasts might still own the potential to form filamentous structure in presence of a correct mating partner. However, this remains rather speculative because the general question of how the ustilaginean yeasts actually relate to pathogenic strains remains to be solved. We added an additional clarification in subsection “Genetic characterization of *MbA”.*

– It is unclear how the bacterial SynCom was constructed and which isolates it contains.

We apologize for not having included the Syn Com isolates list. We have added this to the supplementary now and have explained how the syncom was assembled: For this SynCom we had isolated bacteria from different sites close to Tuebingen in Germany and have sequenced partial 16S to identify taxonomy. Based on these isolates and how often we have found the bacteria on different sites (at least 85% of all sites), we assembled a community with maximum phylogenetic diversity. We have further isolated bacteria from spores of our unsterile *A. laibachii* Nc14 isolate and have added those to cover more phylogenetic diversity.

The detailed information is now available in the new Supplementary file 1.

– The quantification of A. laibachii infection seems kind of rough and prone to subjectivity. An additional, more quantitative method should be used for quantification, for example by qPCR using an amplicon from A. laibachii DNA normalized to a plant amplicon.

We thank this reviewer for pointing this out. Our quantification method is based on the percentage of infected leaves vs non-infected leaves in each individual *A. thaliana* seedling. This method of counting pustules on leaves has been intensely used for evaluating rust fungal infections. As leaves of sterile Arabidopsis plants are small, there are no more than one to two pustules per leaf. Using this range would not give enough resolution to see significant differences and therefore we decided to analyze significant more plants and leaves and focus on presence and absence. For statistical analysis, infection percentage of each seedling is subjected to analysis of variance (ANOVA) for pairwise comparison of the conditions, with Tukey’s HSD test to determine significant differences among them (*P* values <0.05). We performed the experiments in six independent biological replicates with 12 technical replicates and observed significant difference between the *Albugo* treated seedlings and *Albugo* + *Moeziomyces* treated ones. Hence, we are convinced that the data shown in our study is robust and statistically significant.

However, based on the suggestion of this reviewer, we performed qPCR-based relative quantification of the *Albugo* biomass in *A. thaliana* in our new experiments where samples were treated with GH25 hydrolase and Mutant_GH25, respectively (see new Figure 8D). These experiments provide direct evidence for the biological activity of the GH25 hydrolase: *Albugo* biomass was reduced to about 50% in GH25 treatments compared to Mutant GH25 or untreated *Albugo-*infections. This new data is now included in the Results section and the Materials and methods section.

– It is unclear whether GH25 hydrolase suppresses infection of plant leaves by *A. laibachii* or whether it helps *M. albugensis* to parasitize A. laibachii by other means. The authors should examine the effect of isolated GH25 hydrolase on *A. laibachii* infection.

This is a very good question, and we think it is answered by the new data which is already mentioned in the point above: the plant infection experiments with the recombinant GH25 hydrolase show that it directly interferes with Albugo infection (see new Figure 8D). Based on this finding, our next question for upcoming work will be to elaborate the molecular mechanism by which the enzyme inhibits *Albugo*.

– At which stage of infection is *A. laibachii* blocked by GH25 hydrolase or *M. albugensis*?

We observed a slight but not significant decrease in zoosporangial germination efficiency on treatment with the GH25 hydrolase (new Figure 7—figure supplement 5) but found that plant colonization of Albugo is significantly reduced. This is also in line with the microscopic of *A. thaliana* leaves where colonization of *Albugo* is visualized by Trypan blue staining (new Figure 1—figure supplement 2). This shows that in presence of *MbA*, there is little to no colonization of *Albugo* in the leaf surface and only short hyphae are formed. We have also provided more information in the Results.

Reviewer #2:The study by Eitzen et al., characterizes a yeast species isolated from *Arabidopsis* phyllosphere classified as *Moesziomyces albugensis* (Ma). The report provides phylogenetic classification, genome analysis and functional analysis of Ma inhibitory activity towards the white rust oomycete *Albugo laibachii* (Al). Understanding functional interaction in the plant microbiota is a field of intense research with a great fundamental interest, as well as promising applied perspectives regarding the identification of novel biocontrol agents such as potentially Ma. I found the manuscript very pleasant to read and well structured. The work appeared rigorous and conclusions are appropriately supported by the available data. A number of biocontrol yeasts have been isolated in the past, but in my opinion this study stands out by providing direct genetic evidence of the anti-rust activity of Ma, through the identification of one underlying gene (encoding a putative GH25 enzyme) and the characterization of the corresponding mutant phenotype. The Ma genome sequence and gene expression data also provide an interesting resource for further analysis of the biology of this phyllosphere fungus.1) I would have liked to see the activity of the GH25 characterized further. I am lacking a clear hint towards GH25 molecular activity on Al and possibly other members of the phyllosphere.

To address this point, we expressed the GH25 hydrolase (and an enzymatic inactive version) *in Pichia pastoris,* purified the proteins by Ni-NTA affinity chromatography and performed lysozyme activity assays using the recombinant proteins. This showed that the GH25 has a lysozyme activity, which is abolished by a single amino acid exchange in the predicted active site. This data is now included in the manuscript (New Figure 8C) and is described in the Results and the Materials and methods section.

Does ectopic (eg in plants) expression of GH25 has inhibitory effect on Albugo?

To address the question whether the GH25 lysozyme has a direct effect on *Albugo*, we used the recombinant protein. As mentioned above, we did not observe a significant effect on *Albugo* zoospore formation (New Figure 7—figure supplement 5). On the other hand, we found that treatment of *Albugo*-infected Arabidopsis with the active (but not with the inactive) GH25 significantly reduced *Albugo* growth (New Figure 8D). This provides direct evidence showing that the GH25 lysozyme is actually inhibiting infection of Arabidopsis by *Albugo*.

Is there any indication on which species would this enzyme act? Is it acting specifically on Albugo?

In this study we were focusing on the *Moesziomyces – Albugo* interaction on Arabidopsis. However, as also suggested by reviewer 1, we believe that *Moesziomyces* and the GH25 could have a potential used for biocontrol in economically more relevant systems. We will therefore test its impact on *Phytophthora infestans* in future experiments.

What protects Ma from its own secreted hydrolase?

The most likely explanation is that the fungal cell wall does not contain a substrate of the GH25 lysozyme. We are not aware of any other protection mechanism in *Moesziomyces*.

How conserved in GH25 across fungi? Is there anything specific about its sequence (and predicted activity) as compared to other GH25s? to other glycosyde hydrolases?

The GH25 hydrolase encoded by gene *g2490* is predicted has a lysozyme activity, which is essentially an enzyme that cleaves peptidoglycan in bacterial cell walls by catalyzing the hydrolysis of β‐(1,4) linkages between the NAM and NAG saccharides The active site of the GH25 hydrolase is found to contain a DXE motif which is found to be conserved between a wide range of fungi, ranging from other basidiomycetes, ascomycetes and even Chytrids (see also new Figure 7—figure supplement 4). We have additionally attached the raw file for PRALINE multiple sequence analysis for conserved sites.

To what extent is GH25 activity beneficial to Ma?

This is an interesting question, which we cannot answer based on experimental evidence so far. What we show is that the GH25 is required and responsible for the antagonism towards Albugo. One now could hypothesize that this antagonism itself is beneficial for Moesziomyces, i.e. for interspecific competition in niche differentiation. A final answer for this question, however, will be subject of future research.

Could the growth of this fungus be measured on Albugo-infected vs Albugo-free plants? in vitro with or without Albugo?

This is an interesting thought, which will definitely be part of future analysis on the ecological role of the GH25 for *Moesziomyces* as a member of the leaf phyllosphere, particularly in presence of Albugo. Regarding “*in-vitro*” growth, this is experimentally difficult, since Albugo is an obligate biotroph which cannot be propagated in axenic culture.

2) The description of statistical analyses needs attention. I particular, I could not find information on how gene expression and differentially expressed genes were determined.

How was gene expression level determined? How were differentially expressed genes identified? Besides, some information on RNAseq QC and bias correction is needed.

We have added additional, missing. Each file was checked for quality control of RNA-Seq reads prior to analysis with FastQC. Reads were mapped to the *Moesziomyces* and *Arabidopsis* genome with Tophat2 and a count table was generated for *Moesziomyces* counts with HT-seq count. DE-genes were determined with using the "limma"-package in R on "voom"-transformed count data and sample-specific weighting to reduce variance between samples (see Figure 6—figure supplement 1). Counts lower than 10 in average have been extracted. Differentially expressed genes were counted with a threshold of False discovery rate of 0.05 and log2FC > 0. We have the added additional information, in Results section.

Reviewer #3:The manuscript deals with the characterization of a phyllosphere yeast that is supposed to have an antagonistic effect on *Albugo laibachii*.While the manuscript is well written, it suffers from numerous cases of overinterpretation of the results and methodological flaws, as outlined below.Abstract. It is beyond reason that the authors call their yeast isolate "*Moesziomyces albugensis*". First, there is not even a remotely valid description of the "species", and second, it is quite obvious that after checking some of the recent literature on *Moesziomyces* (part of which two of the authors even co-authored) that the isolate they are investigating is clearly *Moeziomyces bullatus*. This needs to be changed throughout the manuscript.

We accept the criticism by the reviewer. We therefore describe the characterized isolate as *Moesziomyces bullatus.* Nevertheless, it needs to be specified that the characterized strain is not a pathogenic strain of *Moesziomyces bullatus,* which is a pathogen of millet. Despite multiple attempts, we were neither able to mate the strain with any isolate of an available collection of pathogenic *Moesziomyces bullatus*, nor have we been able to infect a potential host plant. We have been trying to infect millet plants also with the self-compatible CB1 strain, which is able to form appressoria. However, also this strain did not show invasive growth in any of our experiments (data not shown). Therefore, to specify the isolate that is characterized in this study, we describe it as *Moesziomyces bullatus* ex *Albugo* on *Arabidopsis* (*MbA* for simplicity) throughout the manuscript. This classifies the strain as a member of the species *Moesziomyces bullatus* and at the same time it reflects that it has actually been isolated from Albugo on Arabidopsis plants and not as a pathogen of millet.

Abstract. That the isolates "prevents" Albugo infections is insufficiently shown (but see later criticism on this point).

We realized that “prevent” is not an appropriate term here and rephrased it to “reduces infection”, which underlines that we are observing a quantitative effect and not a complete prevention of *Albugo* infection.

Abstract. The evidence for the claim that GH25 is a major effector of the antagonism is weak, at best.

We accept this criticism, which is also shared by the other reviewers and have performed a series of additional experiments which significantly strengthens this part of the study and provides direct evidence for GH25 being an effector in the observed antagonism. As described above, we produced recombinant protein for the GH25 hydrolase and showed its enzymatic activity. Most importantly, we show that enzymatically active GH25 significantly reduces infection of Arabidopsis by Albugo, which we quantified using qPCR. The additional evidence is shown in the new Figure 8 and in the new Figure 1—figure supplement 2 and Figure 7—figure supplement 2. Detailed descriptions are provided in Materials and methods section and Results of the revised manuscript.

Introduction. This statement is not correct. there are several studies, e.g. by the Macia-Vicente group that have shown that the local environment drives the diversity of root-associated microbiomes.

We agree and have modified the sentence accordingly.

Introduction. the genus Pseudozyma is obsolete. This should be termed Moesziomyces. And the species is known as well, it is *Moesziomyces bullatus*.

We have mentioned the term *Pseudozyma* to introduce the anamorphic yeasts in the manuscript and report biocontrol activity of those group yeasts which have been commonly termed “*Pseudozyma*” in a substantial body of literature. In accordance to the corrected phylogeny, we mentioned the transfer of the genus *Pseudozyma* to *Moesziomyce*s soon after in the Introduction.

Introduction. I was surprised about the statement and have the read Kurse et al., manuscript. In fact, the authors report the opposite, an infrequent sexual reproduction on grass hosts as the most probable explanation for the retainment of a sexual reproduction.

We modified the sentence to avoid confusion (Introduction).

Introduction. I really wonder why the authors cite this less comprehensive phylogeny, while two of them have shown, based on a broader sampling that in fact the species mentioned are embedded within *Moesziomyces* and that *Pseudozyma aphidis* is conspecific with *Moesziomyces bullatus*.

The corrected phylogeny has now been incorporated.

Introduction. The transfer of these species into Pseudozyma had already been done, thus, "can be renamed" is not a correct statement.

We have modified the sentence.

Introduction. This should read *Moesziomyces bullatus* or "A strain formerly classified as *Pseudozyma aphidis* (now *Moesziomyces bullatus*)".

We have changed this as described above.

Introduction. There is no such species name as *Moesziomyces albuginensis*. And this also would not make phylogenetic or evolutionary sense, as the *Moesziomyces bullatus* strain isolated from *Arabidopsis thaliana* is firmly embedded within *Moesziomyces bullatus* and there is no evidence that is should be considered as a biologically distinct species. Also, the evidence of disease "prevention" are at most anecdotal (see later criticism).

We have addressed this point as described above.

Results. This statement alone refutes the previous statement in the Introduction. If the species was isolated from and found to be closely associated with the spores of Albugo, even over years of continuous cultivation cycles, how can the authors confidently claim that the species prevents infection?

The fungal isolate used in this study in particular has been found associated with *Albugo*-infected *A. thaliana* leaf surface. Previous study (Agler et al., 2016) identified this strain of *Moesziomyces* as being a hub microbe in the *A. thaliana* phyllosphere. In this study we provide direct, quantitative evidence that it antagonizes the infection of Arabidopsis by the biotrophic oomycete pathogen *Albugo laibachii* via the GH25 hydrolase. However, we do not want to claim that *Moesziomyces sp.* in general has such a function (to justify this claim, one basically would need to test every fungal isolate individually on the experimental level), and therefore we specified the findings to the individual isolate which we have studied in detail in this study.

Introduction. There are many strains of Moesziomyces bullatus (as Pseudozyma aphidis) that are marked as biocontrol agent. The new strain would just be an addition to that, there is little novelty in this.

We agree with the reviewer that there are several reports on biocontrol activity by *Pseudozyma* sp., which is also addressed in the introduction part of our manuscript. However, we are convinced that our study is not a mere description of an organism with biocontrol activity. An important aspect is on the multitrophic interactions in a phyllosphere microbial community, where *MbA* appears to be regulating the presence of several other microbes in the network. Another novelty of our approach lies in the fact that we haven’t externally introduced an organism to combat pathogenicity; since the yeast is found as a member of the microbial community in the natural environment. Most importantly, our RNAseq analysis revealed the transcriptional activation of secretory enzymes in *MbA* when being co-cultivated with Albugo on the plant leaf, and this approach identified the GH25 lysozyme as a key determinant for the observed microbial antagonism. To the best of our knowledge, this study shows for the first direct and functional evidence on the genetic and protein level in a fungal – oomycete antagonism on/in-planta. Thus, the novelty of this study is o on the (i) community aspect and (ii) the mechanistic evidence of the microbial antagonism, rather than in the identification of a new fungal isolate.

Results. This argumentation is impossible to follow. Just because something is isolated from a special location does not make it a new species.

This has been corrected as described above.

Results. I really cannot see the reason for the authors claiming again the identification was just "Pseudozyma sp." while they could show in a previous manuscript [Kruse et al., 2017] that the species is Meosziomyces bullatus.

This has been corrected as described above.

Results. This interaction does not need to be specific. The test against *Ustilago maydis* is also difficult to justify, as that species is largely unrelated to Moesziomyces bullatus. Instead, other species of the genus Moesziomyces, especially additional strains of Moesziomyces bullatus should have been used.

The intention of this experiment was to observe if there is any difference between the well-characterized pathogenic smut *Ustilago maydis* and the *MbA* strain, which acts as a microbial hub in the *A. thaliana* phyllosphere. As suggested by this reviewer, it would be interesting to test other strains of *Moesziomyces bullatus* against the bacterial SynCom and Albugo to test if there is specific adaptation towards microbial interactions in different isolates of the same species. These lines will surely be followed by our future research in this context, where we want to better explore the ecology Ustilaginales yeasts in different environments.

Results. While in general, it is interesting that the yeast showed an antogonistic effect, dose dependency should have been checked. Several species of the Ustulagoinales are known to produce biosurfactants that alter the properties of the plant cuticle. Thus, the effect observed can probably already explained by this. This could be addressed by additional experiments with a variety of Ustilago and Moesziomyces strains and demonstrating that a lower dose of the Albugo-associated strain inhibits the infection.

We agree that Ustilaginales are known to produce biosurfactants and we would not like to exclude that these secondary metabolites have significant impact on the microbial interactions on the leaf surface. The focus and aim of this study, however, was to investigate possible functions of secreted proteins as potential microbe-microbe effectors. Thus, we decided to investigate the function of the GH25 in more detail and performed the characterization of the recombinant protein, showing that the active GH25 lysozyme is significantly inhibiting infection of Arabidopsis by *A. laibachii* (new Figure 8C,D).

Subsection “The genome of Moesziomyces albugensis”. This lacks a lot of the usual quality control, such as genome completeness statistics, read mapping specifics, testing of the gene predictions by RNA mapping.

We thank the reviewer for pointing out this lack of information on the RNA seq experiment. Each file was checked for quality control of RNA-Seq reads prior to analysis with FastQC. Reads were mapped to the *Moesziomyces* and *Arabidopsis* genome with Tophat2 and a count table was generated for *Moesziomyces* counts with HT-seq count. DE-genes were determined with using the "limma"-package in R on "voom"-transformed count data and sample-specific weighting to reduce variance between samples (new Figure 6—figure supplement 1). We have added additional information, in the Results section.

Subsection “The genome of Moesziomyces albugensis”. Evidence should be presented that this is not the result of a mis-assembly.

We have added the new Figure 2—figure supplement 1, which shows genomic comparison of *MbA* and *Moesziomyces antarctica* T-34 and confirms proper assembly..

Subsection “The genome of Moesziomyces albugensis”. Even though "tempting", this speculation of the authors seems very far-fetched, considering that there are many unrelated groups of the Ustilaginales that cause tumors or not.

We agree that this statement is very speculative and therefore deleted the sentence.

Subsection “The genome of Moesziomyces albugensis”. This part is interesting, but needs to be backed up by RNASeq and metabolic profiling.

Our analysis provided no evidence on upregulation of the secondary metabolite clusters in the RNA seq experiments, hence metabolic profiling was not performed further. Thus, we only speculate on them to be involved in antibacterial activity based on interaction of *MbA* against the SynCom bacterial members on plate (Figure 1—figure supplement 1).

Subsection “The genome of Moesziomyces albugensis”. If the authors reanalyzed that genome, they could probably see that most of the "effector losses" are the result of bad gene calls.

We have not analyzed the *P. flocculosa* genome with respect to gene loss. To avoid misunderstanding, we modified the sentence to make it clearer that we only refer to the statement by Lefebvre et al., (2013), which is apparently not in line with our observation made for MbA.

Subsection “The genome of Moesziomyces albugensis”. I wonder why the authors do not cite or discuss their own findings on several Moesziomyces species previously characterized as non-pathogenic. Ökmen et al., had found years ago that these yeasts possess functional copies of PEP1.

We have added the following in (subsection “The genome of *MbA”)* “*Moesziomyces* sp., possess functional homologues of the *pep1* gene, a core virulence effector of *U. maydis* (Sharma et al., 2019), suggesting that such anamorphic yeasts have the potential to form infectious filamentous structures by means of sexual reproduction (Kruse et al., 2017).”

Subsection “The genome of Moesziomyces albugensis”. The interpretation here cannot be upheld. Given current data, it seems to be a specialty of Ustilago madis that the clusters are expanded (see e.g. Sharma et al., 2014). In addition, the "loss" or presence of the clusters is likely a simple function of phylogenetic distance. Judging from the unfortunately not well resolved phylogeny in [Wang et al., 2015], Kalmanozyma could simply be most unrelated to *Ustilago maydis*.

We agree and have removed the example of *Kalmanozyma brasiliensis*.

Subsection “Genetic characterization of M. albugensis”. This is the first part of the manuscript that seems solid, even though I wonder why non infection trials on the natural host of Moesziomyces bullatus were done.

We appreciate this suggestion by this reviewer. Actually, we tested two different cultivars of millet, *Setaria* sp. and *Echinochloa crus-galli* for susceptibility towards transgenic self-compatible (CB1) strain of *Moesziomyces* sp. However, we could not observe any penetration event in the plant surface, even though the organism was seen to be forming filamentous structure. The same was true for the self-compatible CB1 strain. Thus, we cannot make any statement on a potential role of the *MbA* isolate as a plant pathogen in our paper. On the one hand we were not able to infect a potential host, but on the other hand we also cannot exclude that under certain conditions it could be infectious.

Subsection “Identification of microbe-microbe effector genes by RNA-Seq”. While this cell biology part is again rather solid, the conclusions seem much too strong. For stating that the secretion of GH25 is important, the authors would need to construct a non-secreted version under a native promotor. In addition, the metabolic profile would need to be analysed, both of the secreted metabolites and those retained in the cells. It is conceivable that the GH25 enzyme processes a non-effective precursor to generate the observed effect.

We agree that the statement was quite bolt in the previous version of the manuscript. However, since the experiments with the recombinant GH25 lysozyme (Figure 8) confirmed its role in the interaction, we decided to stay with this phrase, which is now justified in our opinion.

As many results obtained are already discussed in the Results section, the Discussion section is sometimes repetitive. I encourage the authors to keep the Results section free from interpretation and to do this in the Discussion section. As many incorrect claims are made again in the Discussion, I do not comment on them again.

We thank the reviewer for pointing this out. As suggested, we have omitted certain sentences in the Results section to avoid being repetitive.

Subsection “Bioinformatics and computational data analysis”. The information regarding the genome assembly and gene prediction is at best minimal. This needs to much extended, also including the necessary quality checks as indicated earlier.

Again, we agree with the reviewer and have extended this part in the Results section.

Figure 1. This is hardly useful to present the "phylogenetic position" of the strain. There are much more elaborate phylogenies and the authors should simply refer to them. Also, the separation into "apathogenic" and "pathogenic" strains is pointless, as they confirm what was previously known. Likely there are no apathogenic members of the Ustilaginales and the yeasts have just by chance been isolated first from the environment before the pathogenic stage was discovered.

We discussed this critically and, although we agree with this reviewer that there are more extensive phylogenies published, we still believe that the figure is helpful for the reader as an introduction. However, we agree with this reviewer that the term “apathogenic” is not helpful and therefore this has been changed in the revised figure legend.

[Editors’ note: what follows is the authors’ response to the second round of review.]

Essential revisions:Based on the major comments of all three reviewers and a discussion, we request to address the following points before the manuscript can be accepted by eLife:1) The reviewers found the phylogenetic tree in Figure 1 misleading (see also comments about the tree in the previous version of the manuscript). Since a more comprehensive and correct tree is published in Kruse et al., Figure 1 should be removed from the manuscript.

As suggested, we have removed Figure 1 from the manuscript. We felt it was helpful as an introduction to the topic, but we agree that the information is not necessarily required in this manuscript.

2) The GH25 hydrolase exhibits lysozyme activity, which hydrolyzes β‐(1,4) linkages between the NAM and NAG saccharides. These glucans are usually found on bacterial cell walls. However, to the reviewer's knowledge, the cell wall of oomycetes is composed of β-1,3, and β-1,6 glucans. How does GH25 hydrolase inhibit the oomycete Albugo laibachii? This question or hypotheses should be raised and discussed in the Discussion section.

We fully agree that the observed effect of the GH25 on *A. laibachii* infection is surprising (we were surprised by this finding as well). The GH25 enzymes are widely conserved in the fungal kingdom and have been associated with both fungal hyperparasitism, as well as oomycete-oomycete parasitism in Phytium (Horner et al., 2012).

With regard of a direct effect on the Albugo cell wall, one should mention that there is quite sparse knowledge on the actual cell wall composition and we would actually not exclude that it contains a substrate of the GH25. Although the MbA GH25 obviously has lysozyme activity, a comprehensive biochemical characterization of the fungal GH25 family is missing. Thus, it might very well be that these enzymes have unexpected/unknown substrates in the oomycete cell wall. Next, it is unknown which consequences an eventual hydrolysis of an Albugo cell wall compound might have (e.g. an effect on cell wall integrity, or a modification of the cell wall surface which may lead to a block in signal perception – or even an induction of plant defense). In the revised manuscript we now include a discussion on potential functions of GH25 to give the reader an idea on how it might act on the molecular level.

Lastly, one cannot rule out that the effect on *A. laibachii* might be an indirect one: since oomycetes can be found in association with bacteria (which is a largely unstudied/neglected topic) one could even speculate that the GH25 actually targets Albugo-associated bacteria, which might be necessary for virulence of the oomycete. While such a tempting scenario would be also in line with findings of endosymbiotic bacteria in mycorrhizal fungi (work by P. Bonfante and others), at present this is purely speculative and unless there no evidence pointing towards this direction, we’d prefer to not discuss this aspect in the present manuscript.

3) The Title states that GH25 is a secreted enzyme. Unless the authors include an experiment showing secretion, statements on secretion should be toned down in the title and across the manuscript where applicable. (In several instances the authors have already carefully written about "an enzyme with predicted secretion peptide").

We accept this criticism and removed the word “secreted” from the Title, which now reads “…via a GH25 lysozyme”.